



# Surface heat fluxes at coarse-blocky Murtèl rock glacier (Engadine, eastern Swiss Alps)

Dominik Amschwand[1], Martin Scherler[1,†], Martin Hoelzle[1], Bernhard Krummenacher[2], Anna Haberkorn[2], Christian Kienholz[2], and Hansueli Gubler[3]

[1]Department of Geosciences, University of Fribourg, Fribourg, 1700, Switzerland
[2]GEOTEST AG, Zollikofen/Bern, 3052, Switzerland
[3]Alpug GmbH, Davos-Platz, 7270, Switzerland
[†]deceased, 4 June 2022

**Correspondence:** Dominik Amschwand (dominik.amschwand@unifr.ch)

**Abstract.** We estimate the surface energy balance (SEB) of the Murtèl rock glacier, a seasonally snow-covered permafrost landform with a ventilated coarse-blocky active layer (AL) located in the eastern Swiss Alps. We focus on the parametrisation of the turbulent heat fluxes. Seasonally contrasting atmospheric conditions occur in the Murtèl cirque, with down-slope katabatic jets in winter and a strongly unstable atmosphere over the heated blocky surface in summer. We use a novel comprehensive

sensor array both above ground surface and in the coarse-blocky AL to track the rapid coupling by convective heat and moisture fluxes between the atmosphere, the snow cover and the AL for the time period September 2020–September 2022. The in situ sensor array includes a sonic anemometer for eddy-covariance flux above ground and sub-surface long-wave radiation measurements in a natural cavity between the AL blocks. During the thaw seasons, the measurements suggest an efficient (∼90%) export of the available net radiation by sensible and latent turbulent fluxes, thereby strongly limiting the heat available

for melting ground ice. Turbulent export of heat and moisture drawn from the porous/permeable AL contributes to the well-known insulating effect of the coarse-blocky AL and partly explains the climate resiliency of rock glaciers. This self-cooling capacity is counteracted by an early snow melt-out date, exposing the low-albedo blocky surface to the intense June–July insolation, and reduced evaporative cooling due to exacerbated moisture scarcity in the near-surface AL during dry spells. With climate change, earlier snow melting and increased frequency, duration and intensity of heat waves and droughts are

projected. Regarding the parametrisation of the turbulent fluxes, we successfully estimated the year-round turbulent fluxes using a modified Louis (1979) scheme despite seasonally contrasting atmospheric conditions and closed the monthly SEB within $20 \, \text{W m}^{-2}$, except during the snow melt-out months. Detected sensible turbulent fluxes from nocturnal ventilation processes, although a potentially important ground cooling mechanism, are within our $20 \, \text{W m}^{-2}$ uncertainty, because nighttime wind speeds are low. Wintertime katabatic wind speeds had to be scaled to close the SEB, which hints at the limits of parametrisations

based on the Monin–Obukhov theory in complex mountain terrain and katabatic drainage flows. The present work contributes to the process understanding of the SEB and climate sensitivity of coarse-blocky landforms.





## 1 Introduction

Coarse blocky landforms such as rock glaciers, block fields and talus slopes are covered by a thick clast-supported debris mantle typically $\sim 1-5$ m thick. These 'cold rocky landforms' (Brighenti et al., 2021) exhibit lower ground temperatures

compared with adjacent fine-grained or bedrock areas, a phenomenon referred to as 'undercooling' or 'algific' (Wakonigg, 1996; Delaloye and Lambiel, 2005; Millar et al., 2014). This special ground thermal regime is owed to the interactive effects of energy-exchange processes between the atmosphere and the ground arising from the debris mantle properties (Hoelzle et al., 2001). Large blocks and near-the-surface typically sparse fine materials create a vast, highly connected and thus permeable pore space. Depending on the stability of the air column in the debris mantle controlled by temperature gradients, a variable

part of the near-subsurface debris mantle is ventilated and effectively participates in the turbulent heat exchange with the atmosphere. Heat and moisture stored in the ventilated near-surface sub-layer of the debris mantle are rapidly mobilised and contribute to turbulent fluxes at short ($\leq$hourly) time scales. These non-conductive sub-surface heat transfer processes related to storage and phase changes of heat, vapour, water and ice produce the peculiar micro-climate observed in debris mantles. As a consequence, to calculate the surface energy balance, no unambiguous, clearly defined and fixed interface of energy conversion

('*active surface*' sensu Oke (1987)) separating radiative–convective processes in the atmosphere from (dominantly) diffusive processes in the sub-surface is available (Herz, 2006). These 'active surfaces' do not necessarily coincide for different processes (radiation conversion, wind-forced and buoyancy-driven air convection, interception of precipitation, water flow) or quantities (solar radiation, momentum, water), and therefore, vary in time and are not necessarily at ground surface. On seasonally snow-covered sites, snow controls ground–atmosphere heat exchange processes at the surface by its effects on albedo and thermal

insulation (Hoelzle et al., 2003; Zhang, 2005).

These complex interactions between debris mantle, snow cover and atmosphere demand continuous high-resolution sub-surface measurements in addition to the 'classical' above-surface weather station measurements and ground temperature. However, gathering the necessary measurements of sub-surface parameters is challenging in remote mountain terrain, and only a few comprehensive data sets beyond ground temperatures exist in mountain permafrost, an exception being Rist and

Phillips (2005); Rist (2007). They deployed a heat flux plate, ultrasound probes, conductometer, vapour traps and reflectometer probes to characterise the ground hydro-thermal regime of a steep, permafrost-underlain scree slope. Instead, non-conductive heat transfer processes in block fields, talus slopes and debris cover of glaciers have been inferred from their effect on the ground thermal regime, by means of (high-resolution) temperature measurements (Wakonigg, 1996; Wunder and Möseler, 1996; Humlum, 1997; Harris and Pedersen, 1998; Hoelzle et al., 1999; Gorbunov et al., 2004; Růžička et al., 2012; Wagner

et al., 2019; Petersen et al., 2022), in cases aided by gas/smoke tracer experiments (Popescu et al., 2017) or thermal infrared imaging (Shimokawabe et al., 2015).

These challenges have been described in energy balance studies conducted on the Murtèl rock glacier, situated in a cirque in the Upper Engadine (eastern Swiss Alps). In this permafrost landform, the debris mantle overlies a perennially frozen rock-glacier core, is seasonally frozen and roughly coincides with the thermally defined active layer (AL). We'll use the term

'coarse-blocky AL' throughout the text to point to both material property and thermal state. In a micro-climatological study



by Mittaz et al. (2000), significant deviations from a closed surface energy balance (SEB) of up to $78 \, \mathrm{W \, m^{-2}}$ in winter and $-130 \, \mathrm{W \, m^{-2}}$ in summer were found. These deviations exceed methodological uncertainties. The authors suggested that the apparent heat sink in summer and heat source in winter arose from processes unaccounted for in their calculations due to lack of measurements, namely advective and convective heat transport in the coarse-blocky AL. Scherler et al. (2014) addressed the seasonal SEB imbalance by adopting a volumetric energy balance approach consisting in adding a porous interfacial buffer layer able to store and release heat and including radiative and sensible turbulent heat transfer in the coarse-blocky AL. The integration of additional sub-surface heat transfer mechanisms and storage components reduced the deviations to $26 \, \mathrm{W \, m^{-2}}$ in summer and $-29 \, \mathrm{W \, m^{-2}}$ in winter. The remaining seasonal deviations were interpreted as latent heat effects of freezing and thawing of ice in the AL and at the permafrost table. This agreed well with long-term melt rates derived from photogrammetric and three-dimensional borehole deformation leading to subsidence estimates ($\sim 5 \, \mathrm{cm \, yr^{-1}}$) (Kääb et al., 1998; Müller et al., 2014).

In this work, we estimate the SEB of the Murtèl rock glacier using in situ measurements on and within its ventilated coarse-blocky AL. We revisit the micro-climatological studies by Mittaz et al. (2000) and Scherler et al. (2014) adding new measurements from a novel sensor array. Since the deviations present in these works were largely caused by uncertainties in the turbulent fluxes, here we focus on the parameter and parametrisations required for their calculation. We address two questions, a general one and a technical one. The general question is how large are the individual surface heat fluxes on the Murtèl rock glacier and how are these seasonally distributed? From a quantitative process understanding, we gain insight on how the insulating effect of the coarse-blocky AL works and how the rock glacier, a thermally conditioned permafrost landform, responds to climate change. The technical question is which turbulent flux parametrisations and input parameters are appropriate for the study of a seasonally snow-covered ventilated landform in complex mountain terrain? Difficulties in calculating the turbulent fluxes on Murtèl arise at two spatial scales: the meso-scale relief ($10^2$–$10^3$ m) and sub-landform scale ($10^{-1}$–$10^1$ m). The meso-scale relief ($10^2$–$10^3$ m) modifies wind speeds and atmospheric stability in the Murtèl cirque, which must be accurately reflected by the flux parametrisations. In winter, persistent katabatic winds develop on the snow-covered rock faces and slopes and converge in the cirque. These katabatic jets determine the near-surface wind velocities and the vertical turbulent exchange. In summer, surrounding rock spurs weaken the regional valley wind in the sheltered Murtèl cirque. Thermals rising from the strongly heated debris surface create an unstable atmosphere with comparatively low horizontal wind speeds. We test different stability corrections commonly used for debris-covered glaciers. At sub-landform scale ($10^{-1}$–$10^1$ m), the coarse-blocky AL is ventilated and participates in the convective exchange of momentum, heat and moisture with the atmosphere, unless the ground is covered by a sufficiently thick snow cover. Heat and moisture can be drawn from an interfacial buffer layer coupled with the atmosphere. The *'active surface'* (sensu Oke (1987)) does not necessarily coincide with the ground or snow surface. Where should we appropriately measure the meteorological variables, namely surface temperature and surface humidity, needed to estimate the turbulent fluxes – on the ground/snow surface, or at some depth? We describe these turbulent processes at the interface between atmosphere and uppermost coarse-blocky AL ($\sim 1.5 \, \mathrm{m}$ depth) using in situ wind speed measurements and link these to measurements of eddy-covariance sensible fluxes. We then use the gained process



understanding to define the appropriate 'surface' temperature and humidity for the Bowen and bulk aerodynamic approach to parametrise the turbulent fluxes.

Our work contributes to the quantitative process understanding of surface energy fluxes on a ventilated coarse-blocky land-form situated in a complex mountain terrain. The quantification of individual surface heat fluxes and near-surface storage terms will benefit the modelling of past and present mountain permafrost distribution and will help to anticipate the response of coarse-blocky permafrost landforms to climate change.

## 2 Study site and past research

### 2.1 Rock glacier Murtèl

The studied Murtèl rock glacier (WGS 84: 46°25′47″N, 9°49′15″E; CH1903+/LV95: 2'783'080, 1'144'820; 2620–2700 m asl; Fig. 1) is located in a north-facing periglacial area of Piz Corvatsch in the Upper Engadine (eastern Swiss Alps), a slightly continental rain-shadowed high valley (Fig. 2a). Mean annual air temperature (MAAT) is $-1.7°C$; mean annual precipitation is $\sim 900$ mm (Scherler et al., 2014). This tongue-shaped, single-unit (monomorphic, sensu Frauenfelder and Kääb (2000)), active rock glacier is $\sim 250$ m long and $\sim 150$ m wide, surrounded by steep rock faces, and directly connected to a talus slope (2700–2850 m asl) (Fig. 2b). Crescent-shaped furrows ($\sim 3-5$ m deep) and ridges with steep and, in some places, near-vertical slopes dissect the slightly northnorthwestward-dipping surface ($\sim 10-12°$) and create a pronounced furrow-and-ridge micro-topography in the lowermost part of the rock glacier. The snow cover is thicker and lasts longer in furrows than on ridges, influencing the ground thermal regime at small scale (Bernhard et al., 1998; Keller and Gubler, 1993). The coarse-grained and clast-supported debris mantle is only 1–2 m thick in the colder furrows, while it is 3–5 m thick on the rest of the rock glacier. The ground ice table is accessible in a few places. Characteristic clast size ranges from 0.1 to 2 m edge length, with a few rockfall deposited boulders of $\sim 3-5$ m. Fine material ($\leq$ sand) is virtually absent from most of the surface; its volume fraction increases with depth (inverse grading (Haeberli et al., 2006)). Rain and percolating meltwater quickly disappear and the surface is generally dry. Beneath the debris mantle, roughly coinciding with the thermally defined AL, lies the perennially frozen ice-rich rock glacier core. Drill cores have revealed massive ice with ice content $> 90\%$, although boreholes drilled within $\sim 30$ m distance suggest some lateral small-scale heterogeneity (Vonder Mühll and Haeberli, 1990; Arenson et al., 2010). Surface creep rates are $\sim 10$ cm yr$^{-1}$, show a coherent creep pattern and have been slightly accelerating in the past decade (Noetzli and Pellet, 2023). Water emerges seasonally from several springs or seeps at the foot of the rock glacier front and flows to the rock-glacier forefield of till-veneered bedrock not underlain by permafrost (lower boundary of discontinuous permafrost) (Schneider et al., 2012).

The site is snow covered for 7–9 months annually with a up to $1-2$ m thick snowpack. Persistent katabatic winds (Mittaz et al., 2000) develop in the topographically shaded cirque (no direct insolation in November–February) and redistribute snow from the windswept ridges into the furrows, eroding the snow around large blocks (Bernhard et al., 1998). These are preferential spots for *snow funnels* that form after the first snowfall in early winter. Oscillating airflow, resembling breathing, has been observed in these openings through the snow cover. This airflow allows for some vertical heat exchange to occur between the





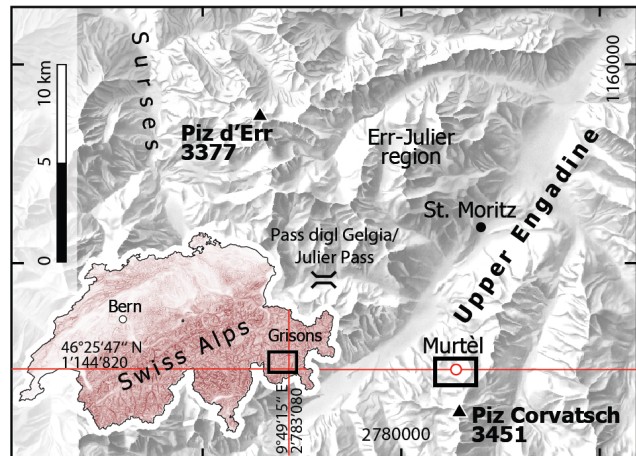

**Figure 1.** Location of Murtèl rock glacier in the Upper Engadine, a high valley in the eastern Swiss Alps. Inset map: Location and extent (black rectangle) of regional map within Switzerland (source: Swiss Federal Office of Topography swisstopo).

coarse-blocky AL and the atmosphere, even during the winter months. As a result, the insulating effect of the snow cover is reduced to some degree, although the extent of this reduction has not been quantified (Bernhard et al., 1998; Keller and Gubler,
125  1993).

## 2.2   Past micro-climatological research

The Murtèl rock glacier and the Murtèl–Chastelets periglacial debris slope have been intensely investigated since ca. 1970. One of the longest continuous mountain permafrost temperature time series (since 1987) worldwide and atmospheric measurements from an automatic weather station (AWS; since 1997) run by the Swiss Permafrost Monitoring Network (PERMOS) have
turned this site into a 'natural laboratory' for mountain permafrost research (summarised by Hoelzle et al. (2002)). Relevant pioneering statistical and process-oriented energy balance studies are the ones by Hoelzle and Haeberli (1995); Hoelzle (1996); Mittaz et al. (2000); Hoelzle et al. (2001); Stocker-Mittaz et al. (2002); Herz et al. (2003); Hoelzle et al. (2003); Hanson and Hoelzle (2004, 2005); Hoelzle and Gruber (2008); Schneider et al. (2012, 2013); Scherler et al. (2014); Wicky and Hauck (2017, 2020); also a series of unpublished diploma (MSc) theses by Stoop (1989); Sutter (1996); Bernhard (1996); Bernhard
et al. (1998); Naguel (1998); Raymond (2001); Oswald (2004); Hasler (2005); Panz (2008); Bircher (2007).

Apart from the two studies on Murtèl by Mittaz et al. (2000) and Scherler et al. (2014) above-mentioned, perhaps the most similar investigation to the present work in terms of ground properties is Isaksen et al. (2003), performed on a ventilated coarse-blocky permafrost site in southern Norway (*Juvvasshøe*); its meso-scale landscape, however, is a wind swept flat mountain top dissimilar to the Murtèl cirque.





## 3 Measurements and data processing

### 3.1 Sensor placement

$50\,\mathrm{m}$ away from the existing Murtèl PERMOS cluster (Noetzli et al. (2019); 'AWS' in Fig. 2a), additional sensors were installed above ground surface and within natural cavities of the porous coarse-blocky AL in August 2020. This PERMA-XT sensor cluster comprised snow and atmospheric sensors above ground surface, AL sensors distributed in natural cavities between the

blocks (Table 1) and two automatic time-lapse cameras in the RGB colour and thermal infrared spectral range. We also used a four-component radiation sensor from the PERMOS cluster for our analysis; its specifications are reported in Scherler et al. (2014); Hoelzle et al. (2022).

The above-ground sensors were located on a rock glacier ridge and included air temperature and humidity sensors, a barometer, a sonic ranger for snow height and a sonic anemometer (CSAT; $3.87\,\mathrm{m}$ above ground surface) for eddy-covariance mea-

surements mounted on a custom-made sensor pylon. On ground level, an unheated tipping-bucket rain gauge measured liquid precipitation. Four unshielded snow thermistors at 0 (ground level), 25, 50 and $100\,\mathrm{cm}$ above ground measured the vertical snow temperature profile. Sensor specifications are presented in Table 1.

The below-ground sensors were distributed in natural cavities at different micro-topographical positions (ridges, slopes and furrows) around the meteo pylon within a $30\,\mathrm{m}$ distance and at different depths beneath ground surface. Five horizontally lying

and vertically hanging thermistor strings and five thermo-anemometers (TP01/WS01) measured temperature and an airflow speed proxy at 5 and $30\,\mathrm{min}$ intervals, respectively. Most sub-surface sensors were concentrated in a $3\,\mathrm{m}$ deep and $0.5-1.5\,\mathrm{m}$ wide instrumented cavity (Fig. 2d). A thermistor string measured the vertical temperature profile of the cavity air (TK1/1–5), complemented by two hygrometers in the near-surface and mid-cavity (HV5; $-0.7$ and $-2\,\mathrm{m}$). Five thermistors (TK6/1–5) were drilled $5\,\mathrm{cm}$ into the blocks at depths corresponding to the TK1 thermistors. Three thermo-anemometers recorded wind

speed at three levels: close to the surface ($-0.35\,\mathrm{m}$), mid-cavity ($-1.5\,\mathrm{m}$) and in a narrow extension at $-2.1\,\mathrm{m}$ (not used in this study). Finally, a back-to-back pair of pyrgeometers mounted at mid-cavity level (CGR3; $-1.55\,\mathrm{m}$) measured the upward and downward long-wave radiation in the cavity. Detailed sensor specifications are presented in Table 1. Since accurate distances were required for the calculation of vertical gradients and fluxes, we triangulated the relative height of the sensors in the instrumented cavity with a laser distance meter/goniometer (Leica DISTO X310).

All sensors were solar-powered and wired to data loggers connected to Internet via the mobile network. Hourly data transmission, when allowed by battery voltage, enabled timely detection of technical failures and intervention. To prevent uncontrolled power shortages during the no-insolation winter period, a protocol progressively sent power-demanding sensors into power-saving mode (50% duty cycle or deactivated completely) as battery voltage decreased. This affected the most power-demanding sonic anemometer and, more rarely, the thermo-anemometer. Power-saving mode was active preferentially during nighttime.

Such incomplete data sets are biased to daytime measurements, and thus daily average might not be representative (Sect. 3.2). For all other sensors, power from diffuse and snow-reflected radiation was sufficient for continuous operation.



**Figure 2.** Sketch map of Murtèl–Corvatsch cirque **(a)** with two active rock glaciers: Murtèl and Marmugnun. **(b)** Wind patterns are seasonally varying in the sheltered cirque. Panels **(c)** and **(d)** show locations of sensors on Murtèl rock glacier and instrumented main cavity, respectively (Table 1). The term 'cavity roof' used throughout the text refers to the uppermost ∼ 1 m of the instrumented cavity.



**Table 1.** PERMA-XT sensor specifications.

| Quantity [unit] | Manufacturer | Sensor type | Accuracy |
|---|---|---|---|
| *Sensors above ground* (atmospheric and snow sensors in Fig. 2c) | | | |
| Air temperature $T_a$ [°C] | CSI[a] | 107 temperature probe[b] | ±0.01 °C |
| Relative humidity (rH for $q_a$) [%] | CSI | HygroVUE10 hygrometer[b] | ±3%; ±0.1°C |
| Barometric pressure $P$ [Pa] | CSI/SETRA | CS100 barometer | ±1.5 hPa |
| Eddy-covariance flux[c] | CSI | CSAT3B 3-D sonic anemometer | |
| Liquid precipitation [mm h$^{-1}$] | CSI | SBS500 tipping bucket rain gauge (unheated) | ±30% (undercatch) |
| Snow temperature [°C] | TE Connectivity[d] | 44031RC NTC thermistors (0, 25, 50, 100 cm a.g.l., unshielded) | ±0.1°C |
| Snow height $H_s$ [cm] | CSI | SR50A sonic ranging sensor | max{±1 cm, ±0.4%} |
| Automatic camera | MOBOTIX | M16B IP camera (RGB) | |
| *Sensors below ground* (active-layer sensors in Fig. 2c) | | | |
| Air temperature $T_a(z)$ [°C] | TE Connectivity[d] | 44031RC NTC thermistor chain TK1/1–5 | ±0.1°C |
| Relative humidity (rH for $q_{sa}$) [%] | CSI | HygroVUE5 hygrometer | ±0.3°C; ≤ ±4% |
| Rock temperature $T_r(z)$ [°C] | TE Connectivity[d] | 44031RC NTC thermistor chain TK6/1–5 (drilled 5 cm into the blocks) | ±0.1°C |
| Long-wave radiation $L_{al}$ [W m$^{-2}$] | Kipp & Zonen | CGR3 pyrgeometer (4.5−42 µm, FoV 150°) | < 4 W m$^{-2}$ |
| Airflow speed proxy $u$ [-] | Hukseflux | TP01 thermal properties sensor (formerly WS01) (artificial leaf used as hot-film anemometer[e]) | ≤ ±25% or ≤ ±0.2 m s$^{-1}$ |

Measurement range and accuracy by manufacturer/vendor. Specifications of PERMOS sensor available in Scherler et al. (2014) and Hoelzle et al. (2022).
[a] CSI: Campbell Scientific, Inc. [b] Sensors in radiation shield RAD10E. [c] CSAT measures 3-dimensional high-frequency sonic wind speed and derives sonic buoyancy flux. [d] Snow temperature setup and thermistor strings manufactured by Waljag GmbH. [e] Semi-quantitative airflow speed derived from measured heat flux.

## 3.2 Data processing

In the present work, we analysed data of two years from 1 September 2020 to 30 September 2022, except for the eddy-covariance data. The sonic anemometer (CSAT) was operational from 5 November 2020.

### 175    3.2.1 Surface radiation

All four components of the radiative heat fluxes at the surface – i.e. incoming and outgoing short-wave ($S$) and long-wave ($L$) radiation – were measured directly by the PERMOS micro-meteorological station with two back-to-back pairs of Kipp&Zonen CM3 pyranometers (0.3−3 µm) and CG3 pyrgeometers (5−50 µm) (Hoelzle et al., 2022). We applied the same radiation corrections as in Hoelzle et al. (2022): (1) instrumental corrections of both pyranometers with nighttime short-wave radiation measurements (pyranometer offset correction), (2) correction for snow cover of the upward-looking pyranometer ($S^{\downarrow} = S^{\uparrow}/0.86$





when $S^\uparrow > S^\downarrow$), (3) Sicart et al. (2005) correction of the incoming long-wave radiation, and (4) rejection of incoming short-wave radiation at low sun angles (if albedo exceeds unity).

### 3.2.2 Eddy-covariance data

We estimated the sensible turbulent flux $Q_H$ using the eddy-covariance method, based on the covariance of vertical wind speed
fluctuations $w'$ and temperature fluctuations $T'$, measured by a CSI CSAT (CSI CSAT3B manual, 2020) and averaged over
30 min (Foken (2017); cf. Sect. 4.2.2):

$$Q_H^{\text{eddy}} = \rho_a C_p \overline{w'T'}, \tag{1}$$

where $(\rho_a C_p)$ refers to the volumetric heat capacity. We calculated buoyancy fluxes from the eddy raw data (10 Hz sampling
rate) using Campbell Scientific's software EasyFlux™ and following the conventional pre-processing chain (Radić et al., 2017;
Fitzpatrick et al., 2017; Steiner et al., 2018). Processing steps were trend and outlier removal, despiking (Vickers and Mahrt,
1997; Foken et al., 2012), coordinate rotation using the double rotation method (Tanner and Thurtell, 1969) and spectral
corrections (Moncrieff et al., 1997; Massman, 2000).

    Due to lack of fast-response moisture measurements (no high-frequency gas analyser), the eddy-covariance method yielded
the sonic *buoyancy flux* $\overline{w'T'_{snc}}$, which, for $|Q_H| > |Q_{LE}|$, is close to but not equal to the sensible heat flux $\overline{w'T'}$ (Liu and
Foken, 2001). We asses the discrepancy using the Liu and Foken (2001) modified Bowen ratio method based on Schotanus
et al. (1983) ('SND correction'):

$$\overline{w'T'} = \overline{w'T'_{snc}} \left( 1 + \frac{0.51 \bar{T}_a C_p}{L_v \text{Bo}} \right)^{-1}, \tag{2}$$

where $T_{snc}$ is the sonic temperature, $\bar{T}_a$ is the air temperature (averaged over 30 min), and Bo refers to the Bowen ratio
(Sect. 4.2.2). $C_p$ and $L_v$ are constants defined in Sect. 4.2.2.

### 3.2.3 Snow height and snow water equivalent

We measured snow height with a sonic distance sensor (Table 1) on a rock-glacier ridge, in addition to the nearby PERMOS
snow height measurements. Raw measurements were compensated for variations of the speed of sound with air temperature,
following the manufacturer's guidelines (CSI SR50A manual, 2020).

    We used the semi-empirical ΔSNOW model (Winkler et al., 2021) to convert the measured snow height to snow water
equivalent (SWE). This parsimonious model requires snow height and its temporal changes as the sole input. Calibration data
used to develop their model were gathered in the Swiss and Austrian Alps, in climatic regions similar to the Engadine.

### 3.2.4 Precipitation

We measured liquid precipitation with an unheated tipping bucket rain gauge (Table 1) mounted on a rock-glacier ridge.

    Snow or sleet (a mix of snow and rain) can fall in any month of the year. The distinction from rain is important because of
the latent heat of melting $L_m$ that ice carries and by far exceeds the sensible heat of rainwater ($c_w \Delta T$). In the comparatively





dry climate of the Engadine, wet-bulb temperature can be a better discriminator than dry-bulb temperature (Froidurot et al., 2014). Wet-bulb temperature $T_{wb}$ [K] is defined in (Stull, 2017):

$$e - e^*(T_{wb}) = -\gamma(T - T_{wb}) \quad [\text{Pa}] \tag{3}$$

where $\gamma = (c_p P)/(0.622 L_v)$ [Pa K$^{-1}$] is the psychrometric 'constant'. Eq. 3 is solved iteratively. Based on hourly images from the automatic time-lapse camera, we defined a conservative wet-bulb temperature threshold of $2°$C that separates snow or sleet ($T_{wb} < 2°$C) from rain.

### 3.2.5 Sub-surface long-wave radiation

We corrected raw outputs $L_{raw}$ from the two pyrgeometers in the instrumented cavity by accounting for the long-wave radiation emitted by the instruments themselves (Kipp & Zonen CGR3 manual, 2014):

$$L = L_{raw} + \sigma T^4_{\text{CGR3}}, \tag{4}$$

where $T_{\text{CGR3}}$ refers to the pyrgeometer housing temperature. Large ($> 0.5°$C) or rapid changes in housing temperature differences between the back-to-back mounted pyrgeometer indicate dust or water deposition on the upward-facing pyrgeometer window. Such disturbed measurements appeared in the high-resolution (10 min) data, but did not significantly affect the daily net long-wave radiation balance in the sheltered cavity.

### 3.2.6 Sub-surface airflow speed

We refer to the sub-surface 'wind' in the cavity as 'airflow' to differentiate it from the atmospheric wind. We used the Hukseflux WS01/TP01 sensor to perform airflow speed measurements in the AL cavities. This sensor included a heated foil that measures a cooling rate expressed as a convective heat transfer coefficient $h_{\text{WS}}$ [W m$^{-2}$ K$^{-1}$] related to airflow speed (Hukseflux WS01 manual, 2006). We did not convert $h_{\text{WS}}$ to airflow speed but process this variable as a semi-quantitative indicator assuming $u \propto h_{\text{WS}}$, sufficient for our purpose. WS01/TP01 do not resolve the direction of the airflow (hence the term 'speed' instead of 'velocity'). Deposition and evaporation of liquid water can disturb measurements ($h_{\text{WS}}$ increase), as revealed by wrapping the heated foils in moist tissues. WS01 measurements during precipitation events were filtered out. Repeated zero-point checks were performed throughout the snow-free season by enclosing the heated foil in small and dry plastic bags for a few hours, ensuring stagnant conditions with zero airflow speed. Neither drift nor temperature dependency beyond measurement uncertainty was detected.

## 4 Surface energy balance calculation

### 4.1 Energy balance of Murtèl near-surface AL

The point-scale surface energy balance at seasonally snow-covered sites accounts for net radiation $Q^*$ [Wm$^{-2}$], composed of short- and long-wave radiation components; turbulent fluxes, composed of sensible heat $Q_H$ [Wm$^{-2}$] and latent heat $Q_{LE}$





$[\mathrm{Wm^{-2}}]$; melt energy of snow $Q_M$ at the surface $[\mathrm{Wm^{-2}}]$; energy from precipitation $Q_P$ $[\mathrm{Wm^{-2}}]$ and heat flux $Q_G$ $[\mathrm{Wm^{-2}}]$ into/from the ground when snow free, replaced by conductive heat flux across the snow cover $Q_S$ $[\mathrm{Wm^{-2}}]$ when snow covered:

$$\underbrace{S^{\downarrow}+S^{\uparrow}+L^{\downarrow}+L^{\uparrow}}_{\text{net radiation } Q^*}+\underbrace{Q_H+Q_{LE}}_{\text{turbulent fluxes}}+Q_M+Q_P+Q_{G/S}=0 \quad [\mathrm{W\,m^{-2}}]. \tag{5}$$

Fluxes are counted as positive if these provide energy to the reference surface, i.e. the terrain or snow surface. Unlike Scherler

et al. (2014), we consider the reference surface as an infinitely thin skin layer without storage. Fluxes must be balanced at all times (Eq. 5). Ground $Q_G$ and snow $Q_S$ heat fluxes measured beneath the surface are extrapolated to the reference surface by means of the calorimetric correction.

## 4.2 Flux parametrisations

We estimated the fluxes (terms in Eq. 5) as follows: all radiative fluxes were derived from on-site measurements, net radiation

$Q^*$ both above the surface (PERMOS data) and within the AL (PERMA-XT data); turbulent fluxes $Q_H$ and $Q_{LE}$ were estimated using the Bowen energy balance and the bulk aerodynamic methods and directly from eddy-covariance measurements. Snowmelt $Q_M$ and precipitation $Q_P$ heat fluxes were estimated using the calorimetric method from SWE estimates and on-site measured rainfall rates, respectively. The heat flux in the snowpack $Q_S$ is the calorimetrically corrected conductive heat flux at the base of the snowpack. The ground heat flux $Q_G$ (flux in the near-surface AL) was estimated analogously to the

calorimetrically corrected net long-wave radiation measured in situ in the AL at $1.5\,\mathrm{m}$ depth.

### 4.2.1 Surface radiative fluxes

The net radiation $Q^*$ is the sum of the measured and corrected radiation components (Sect. 3.2.1):

$$Q^* = \underbrace{S^{\downarrow}+S^{\uparrow}}_{S^*}+\underbrace{L^{\downarrow}+L^{\uparrow}}_{L^*}, \tag{6}$$

where $S^{\uparrow}$ is the outgoing short-wave radiation, $S^{\downarrow}$ is the incoming short-wave radiation, and net shortwave radiation $S^*$ is the

sum of both; correspondingly, $L^{\uparrow}$ represents the outgoing long-wave radiation, $L^{\downarrow}$ is the incoming long-wave radiation, and net long-wave radiation $L^*$ is the sum of both.

### 4.2.2 Surface turbulent heat fluxes

We estimated the turbulent flux using three different methods: (1) the Bowen energy balance method (Bowen, 1926), (2) the bulk aerodynamic method (Mittaz et al., 2000; Hoelzle et al., 2022) and (3) directly with the eddy-covariance method from

CSAT measurements (in the lack of a fast-response vapour analyser, only the sensible turbulent flux $Q_H$ is estimated; Sect. 3.2.2).



**Bowen energy balance method**

The Bowen ratio Bo is defined as the ratio of sensible to latent heat flux (Bowen, 1926; Ohmura, 1982; Oke, 1987) and reflects the partitioning of the turbulent fluxes into sensible and latent components:

$$\text{Bo} := \frac{Q_H}{Q_{LE}} = \frac{K_h}{K_w}\frac{C_p\Delta T}{L_v\Delta q} \approx \frac{C_p\,(T_a - T_s)}{L_v\,(q_a - q_s)} \tag{7}$$

where $K_h$ represents the eddy diffusivities for sensible heat and $K_w$ is water vapour. Invoking the similarity principle and assuming $K_h \approx K_w$ (Ohmura, 1982), the Bowen ratio can be calculated from the isobaric specific heat capacity of air [$C_p = C_{pd}(1 + 0.84q$ with $C_{pd} = 1005\,\text{J kg}^{-1}\,\text{K}^{-1}$] (Reid and Brock, 2010; Hoelzle et al., 2022), the latent heat of vaporisation $L_v$ (if $T_s \geq 0°\text{C}$) or sublimation $L_s$ (if $T_s < 0°\text{C}$) [$2.48\cdot10^6\,\text{J kg}^{-1}$, $2.83\cdot10^6\,\text{J kg}^{-1}$] and the gradients of temperature $\Delta T$ and specific humidity $\Delta q$ (specified below). The sensible and latent turbulent fluxes are then expressed as a function of the available

energy $Q^* + Q_M + Q_{G/S}$:

$$Q_H^{\text{Bo}} = \frac{Q^* + Q_M - Q_{G/S}}{1 + 1/\text{Bo}}, \quad Q_{LE}^{\text{Bo}} = \frac{Q^* + Q_M - Q_{G/S}}{1 + \text{Bo}}. \tag{8}$$

The ground and snow surface temperature $T_s$ [°C] was calculated from the measured long-wave radiation components $L$ and the Stefan–Boltzmann law via (Oke, 1987):

$$\varepsilon\sigma T_s^4 = L^\uparrow - (1 - \varepsilon)L^\downarrow, \tag{9}$$

where $\varepsilon$ is the surface emissivity, $\sigma$ represents the Stefan–Boltzmann constant ($\sigma = 5.670 \times 10^{-8}\,\text{W m}^{-2}\,\text{K}^{-4}$) and $L^\uparrow$ and $L^\downarrow$ represent the measured outgoing and incoming long-wave radiation, respectively. We took the emissivity value of $\varepsilon = 0.96$ from Scherler et al. (2014).

The surface specific humidity $q_s$ [g g$^{-1}$] is either the specific humidity of the air in the near-surface coarse-blocky AL $q_{sa}$ (measured) or the saturated snow surface $q_{ss}^*(T_s, P_a)$, depending on whether or not the snow cover is thick enough to suppress

convective air exchange between the AL and the atmosphere (Sect. 6.3.1):

$$q_s = \begin{cases} q_{sa}, & \text{for } H_s \leq {}^{\text{SEB}}H_s^{\text{crit}} \text{ (snow free or open snow cover)} \\ q_{ss}^*, & \text{for } H_s > {}^{\text{SEB}}H_s^{\text{crit}} \text{ (decoupling snow cover).} \end{cases} \tag{10}$$

The critical snow height ${}^{\text{SEB}}H_s^{\text{crit}}$ is reached when the AL is so strongly decoupled from the atmosphere that the convective fluxes across the snow cover are no longer detectable by our SEB estimations. We determine the snow height ${}^{\text{SEB}}H_s^{\text{crit}}$ using the AL air temperature, specific humidity and airflow speeds (Sect. 6.3.1). The specific humidity of the air $q_a$ was calculated

from the measured air temperature $T_a$ and relative humidity at the measurement level $z_m$ of 2 m above ground surface.





**Bulk aerodynamic method**

In the bulk parametrisation, sensible $Q_H^{\text{bulk}}$ and latent $Q_{LE}^{\text{bulk}}$ turbulent fluxes are driven by the gradients of temperature $\Delta T = T_a - T_s$ [K] and specific humidity $\Delta q = q_a - q_s$ [g g$^{-1}$], respectively, and horizontal wind speed $u$ [m s$^{-1}$]:

$$Q_H^{\text{bulk}} = \rho_a C_p u (T_a - T_s) C_{bt} \tag{11}$$

$$Q_{LE}^{\text{bulk}} = \rho_a L_{\{v,s\}} u (q_a - q_s) C_{bt} \tag{12}$$


where $\rho_a$ [kg m$^{-3}$] represents air density. The flux–gradient relationship is modified by atmospheric stability, accounting for enhanced turbulent fluxes in an unstable atmosphere and suppressed turbulent fluxes in a stable atmosphere, by means of the bulk exchange factors $C_{bt}$ (bulk turbulent heat and vapour transfer coefficients) [unitless, $-$]. Different formulations of the stability functions exist. Here, we use the modified Louis (1979) scheme (Song, 1998), motivated by its use in the GEOtop

model, a distributed hydrological model designed for complex terrain (Rigon et al., 2006; Endrizzi et al., 2014), and other studies (e.g. Essery and Etchevers, 2004). We compare the modified Louis (1979) scheme with the iterative Monin–Obukhov scheme and the widely used Businger–Dyer scheme (Oke, 1987). The latter has been used in the previous SEB studies on Murtèl by Mittaz et al. (2000); Scherler et al. (2014) and in many SEB estimates on debris-covered glaciers (Brock et al., 2010; Reid and Brock, 2010; Reid et al., 2012; Steiner et al., 2018, 2021), despite discrepancies (overestimated fluxes) at

strongly unstable atmosphere noted by Steiner et al. (2018). The issue with the Businger–Dyer parametrisation is that near-surface wind speeds on topographically sheltered rough terrain tend to be lower for a given atmospheric stability compared with the conditions under which the empirical Businger–Dyer parametrisation was originally developed (flatland in Kansas, USA (Businger et al., 1971; Haugen et al., 1971)). This approach is therefore problematic in complex terrain, including our study site. Finally, we test the parametrisation without stability correction ('bulk c0'), which corresponds to the special case of

a neutral atmosphere. Detailed explanations of the parametrisation schemes of the turbulent heat transfers are described in the appendix (Appendix Sect. A).

Accurate estimates of turbulent fluxes rely on representative values for the roughness lengths for momentum $z_{0m}$, heat $z_{0h}$ and moisture $z_{0q}$ (Steiner et al., 2018; Smeets et al., 1999; Smith, 2014). We calculated the roughness length for momentum $z_{0m}$ from the eddy-covariance data following Conway and Cullen (2013); Fitzpatrick et al. (2017):

$$z_{0m} = (z_u - d) \exp \left\{ -k_* \frac{\bar{u}_h}{u_*} - \psi_m \left( \frac{z_u - d}{L} \right) \right\} \tag{13}$$

where $\bar{u}_h$ [m s$^{-1}$] represents the mean horizontal wind speed, $u_*$ [m s$^{-1}$] is the friction velocity, $k_*$ [0.40] is the von Kármán constant, $z_u$ [m] is the measurement height of wind speed, $L$ [m] is the Obukhov length, $\psi$ is the integrated stability function (Appendix Sect. A) and $d$ is the zero-plane displacement height. We set the (unknown) zero-plane displacement height $d$ to zero (Miles et al., 2017). The sensor height $z_u(t)$ is the variable distance above the ground or snow surface. The recommended

filtering is to use only $\bar{u}_h > 3$ m s$^{-1}$ under near-neutral conditions $|z_u/L| < 0.05$ (Radić et al., 2017; Nicholson and Stiperski, 2020). We compared our eddy-covariance-derived roughness length for momentum $z_{0m}$ with the aerodynamically derived values reported in Mittaz et al. (2000). For simplicity, the scalar lengths for heat $z_{0h}$ and humidity $z_{0q}$ are often considered



equal to the momentum roughness length $z_{0m}$ (Sicart et al., 2005; Reid and Brock, 2010), or 1–3 orders of magnitudes smaller (Smeets and van den Broeke, 2008). Here, we could not independently estimate the scalar roughness lengths $z_{0h}$ and $z_{0q}$ due

to lack of humidity-corrected sonic temperature or high-frequency humidity measurements. We assumed an equal roughness length for heat and moisture, $z_{0h} = z_{0q}$, and used the unknown ratio $\hat{R}_{z0} := z_{0h}/z_{0m} = z_{0q}/z_{0m}$ as a calibration parameter.

Finally, we compared the sensible turbulent flux with Oerlemans and Grisogono (2002)'s katabatic model specifically developed for conditions of katabatic or nocturnal drainage flows. This is different from all the above-mentioned parametrisations as this predicts a quadratic rather than near-linear relation between $Q_H$ and the driving temperature difference $\Delta T$. Since we

lacked vertical profile observations of potential temperature required for this parametrisation, we could test this bulk method. Instead, we checked the validity of the quadratic $Q_H$–$\Delta T$ relation on Murtèl (cf. Radić et al., 2017) and show wind profile measurements collected by Mittaz et al. (2000) in the years 1997–2000 (Appendix Sect. D).

### 4.2.3 Snow melt energy, snow heat flux and snowpack sensible heat storage changes

Snow melt energy $Q_M$ is absorbed by the melting snowpack. We estimated it from the daily change in SWE [kg m⁻²] derived

from the snow height data $H_s$ (sonic ranger data) with the semi-empirical parsimonious ΔSNOW model (Winkler et al., 2021):

$$Q_M \approx L_m \frac{\Delta(\text{SWE})}{\Delta t}, \quad \text{if } T_b = 0°\text{C}, \tag{14}$$

where $L_m$ [336 kJ kg⁻¹] represents the latent heat of fusion for ice. We considered runoff-generating snow melting in spring occurring when temperature at the base of the snowpack $T_b$ reaches the melting point and ignored melting–refreezing events

within the snowpack.

We calculated the conductive heat flux across the single-layer snowpack $\tilde{Q}_S$ ignoring transient effects (Mittaz et al., 2000):

$$\tilde{Q}_S = -k_s \frac{dT}{dz} \approx -k_s \frac{T_s - T_b}{H_s} \approx -k_s \frac{T_{25} - T_0}{\Delta z}, \quad \text{if } H_s > 30 \text{ cm and 0 otherwise}, \tag{15}$$

where $dT/dz$ is a linearised temperature gradient in the lowermost 25 cm of the snowpack. To ensure that both thermistors are snow covered, the minimum snow height $H_s$ required is 30 cm. We related the snow thermal conductivity $k_s$ to the snow density

via the empirical equation $k_s := k_s(\rho_s) = 2.93 \, (10^{-6}\rho_s^2 + 10^{-2})$ developed for Murtèl (Keller and Gubler, 1993; Hoelzle et al., 2022), where we estimated the bulk density of the snowpack with the ΔSNOW model via $\bar{\rho}_s = \text{SWE}/H_s$ (Winkler et al., 2021).

$\tilde{Q}_S$ was calculated near the base of the snowpack instead of at the snow reference surface to which Eq. 5 refers to. With this approach, we used the more stable snow density and thermal conductivity near the snowpack base, which is less affected by compaction than the near-surface layers that receive fresh snow. The heat flux $\tilde{Q}_S$ was extrapolated to the snow surface by

adding the sensible storage changes in the snowpack $\Delta Q_S$ above $z_s$ ('calorimetric correction') (Foken, 2008; Liebethal et al., 2005; Boike et al., 2003). These changes in cold content were estimated as:

$$-\Delta Q_S = c_s(m_s/A)\Delta T/\Delta t \approx c_s \, \text{SWE} \, \langle\Delta T\rangle/\Delta t \tag{16}$$

where $(m_s/A)$ represents the mass of the snowpack per area, i.e. the SWE [kg m⁻²], $c_s$ [2.050 kJ kg⁻¹ K⁻¹ is the specific heat capacity of snow or ice at 0°C] and $\langle\Delta T\rangle$ [K] represents the layer-averaged snow temperature changes. Warming absorbs





heat and cooling releases heat, hence the minus sign in Eq. 16. The calorimetrically corrected snow heat flux $Q_S$ was then
calculated as:

$$Q_S = \tilde{Q}_S + \Delta Q_S. \tag{17}$$

### 4.2.4 Precipitation heat flux

The rainfall heat flux $Q_P^{rain}$ was estimated via (Sakai et al., 2004; Reid and Brock, 2010):

$$Q_P^{rain} = C_w \dot{W}_{rain} (T_P - T_s), \tag{18}$$

where $C_w = \rho_w c_w$ [4.18 MJ m$^{-3}$ K$^{-1}$] is the water volumetric heat capacity and $\dot{W}_{rain}$ [m$^3$ m$^{-2}$ s$^{-1}$] is the rainfall rate
intercepted at the surface. Precipitation temperature $T_P$ was approximated using the wet-bulb temperature $T_{wb}$, calculated from
air temperature and relative humidity (Eq. 3). We assumed precipitation in form of rain if $T_{wb} \geq 2°$C. Water contributions from
up-slope flowing onto the rock glacier and liquid precipitation falling into the snowpack were not accounted for.

The heat flux $Q_P^{solid}$ due to rapid melting of sleet or shallow summertime snow ($T_{wb} < 2°$C, $H_s < 10$ cm) was roughly
estimated as:

$$Q_P^{solid} \sim L_m \rho_w W_s / \Delta t, \tag{19}$$

where $W_s$ [m$^3$ m$^{-2}$] represents the amount of solid precipitation that melts in the pluviometer in the time period $\Delta t$ [s]. Our
measurement setup was not designed to accurately record the precipitation rate during mixed rain-and-snow fall or the fraction
of ice crystals and liquid water in the total precipitation. The associated heat flux $Q_P^{solid}$ is intended as an order-of-magnitude
estimate.

### 4.2.5 Near-surface ground heat flux and sensible heat storage

Analogously to the calorimetrically corrected snow heat flux $Q_S$, the ground heat flux $Q_G$ is the sum of the net long-wave
radiation in the instrumented cavity $\tilde{Q}_{r_{al}}$ and the sensible storage changes $\Delta Q_{stor}$ of the virtual layer between the ground
surface and the depth of the pyrgeometer:

$$Q_G = \tilde{Q}_{r_{al}} + \Delta Q_{stor}. \tag{20}$$

The in-cavity net long-wave radiation was calculated from the upwards $L_{al}^{\uparrow}$ and downwards $L_{al}^{\downarrow}$ long-wave radiation compo-
nents measured with a back-to-back pyrgeometer pair installed in the instrumented cavity (Table 1):

$$\tilde{Q}_{r_{al}} = (L_{al}^{\uparrow} - |L_{al}^{\downarrow}|), \tag{21}$$

which is not to be confused with the long-wave radiation $L$ measured 2 m above the surface (Sect. 4.2.1).

Rock mass subject to changing temperatures constitutes a heat source or sink (Scherler et al., 2014). The sensible heat storage
change in the (dry) blocks $\Delta Q_{stor}$ is proportional to the rate of change of rock temperature, assuming a constant volumetric





heat capacity $\partial\{(1-\phi_d)\rho_r c_r\}/\partial t = 0$. In the lack of rock temperatures over the 2 year time period analysed in this work, we use in-cavity air temperatures $\bar{T}_a$ averaged over a thermal adjustment time scale $\Delta t_r$ (1 day) as a surrogate (Sect. 6.1.2;

Appendix Sect. C). The rate of sensible heat storage and release of the blocks was then approximately calculated as (Liebethal and Foken, 2007; Ochsner et al., 2007; Brock et al., 2010):

$$-\Delta Q_{stor} = \int\limits_{-z_s}^{0} \frac{\partial}{\partial t}\{(1-\phi_d)\rho_r c_r T_r(z)\}\,\mathrm{d}z \approx (1-\phi_d)\frac{\langle\rho_r c_r\rangle}{\Delta t_r}\sum_i\{\langle\bar{T}_a(z_i,t+\Delta t_r)\rangle - \langle\bar{T}_a(z_i,t)\rangle\}\Delta z_i \tag{22}$$

where $z_s = 155\,\mathrm{cm}$ is the distance from the pyrgeometer pair to the ground surface (reference level), $\rho_r$ is the rock density [2690 kg m$^{-3}$] (Corvatsch granodiorite, Schneider (2014)), $c_r$ is the specific heat capacity [790 J kg$^{-1}$ K$^{-1}$], $\phi_d = 0.4$

(Scherler et al., 2014) is the AL porosity and $T_r(z)$ and $T_a(z)$ are the vertical rock and in-cavity air temperature profile [°C], respectively. In the discretised formulation, temperatures $\langle\bar{T}(z_i)\rangle$ are layer-wise averages in the $i$-th layer with thickness $\Delta z_i$ (denoted by $\langle\cdot\rangle$) derived from the thermistor string TK1/1 and the radiometric surface temperature $T_s$. As in Eq. 16, warming absorbs heat and cooling releases heat, hence the minus sign in Eq. 22.

## 5  Results

### 5.1  Meteorological conditions

The weather in each season differed markedly between the two years analysed in the present work (2020–2022; Fig. 3). The winter 2020–2021 was colder than the 2021–2022 one (November–April: average temperature: $-6.2$°C vs. $-5.3$°C; minimum daily average temperature: $-16.5$°C vs. $-15.1$°C) and richer in terms of snow amount (November–April: average snow height measured on a wind-swept ridge: 76 cm vs. 54 cm) and duration (early onset of snow cover: 5 October vs. 3

November; later melt-out: mid-June vs. mid-May). Summer 2021 was cool-wet compared with the hot-dry summer 2022; temperatures were lower (July–August: average: 6.9°C vs. 9.3°C) with frequent passage of synoptic fronts, often bringing cold air ($\leq 3$°C; minimum daily average temperature: 0.7°C vs. 5.6°C) and mixed precipitation (sleet). Snowfall occurred in a few days throughout the summer and melted within hours. A few snow patches survived over the summer after melt-out of the winter snowpack in mid-June. In contrast, the hot-dry summer 2022 was marked by three heat waves (in June, July and

August) and daily minimum temperatures not below 5°C. Several dry spells occurred during this season; the longest one was an 11-day long dry spell within the 5–19 July heat wave. Almost no precipitation was recorded between 20 June and  1 Aug, despite some convective precipitation events recorded on the nearby *Piz Corvatsch* cable car top station (MeteoSuisse station at 3294 m asl). Discharge data of the rock glacier outflow (own measurements, not shown), camera images and field observations (fresh debris flow deposits, flooding of furrows) revealed rainwater funnelled onto the rock glacier. Data gaps for this period

were filled with MeteoSuisse precipitation data from the nearby station *Piz Corvatsch*.

Wind speed (measured by PERMOS) in the sheltered Murtèl cirque were generally low in (hourly means: $1-3$ m s$^{-1}$; peak wind speed $\sim 5$ m s$^{-1}$; Fig. 2b). The wind pattern was often marked by a strong diurnal cycle. Peak wind speed was reached during the night in winter (strong katabatic wind blowing down-slope from SE) and in the afternoon in summer (regional valley





wind known as the *Maloja wind* blowing from WNW–WSW overruled a local anabatic wind). Summer nights were calm or
with weak katabatic winds (wind speed: $\leq 2 \text{ m s}^{-1}$ from W–SE).

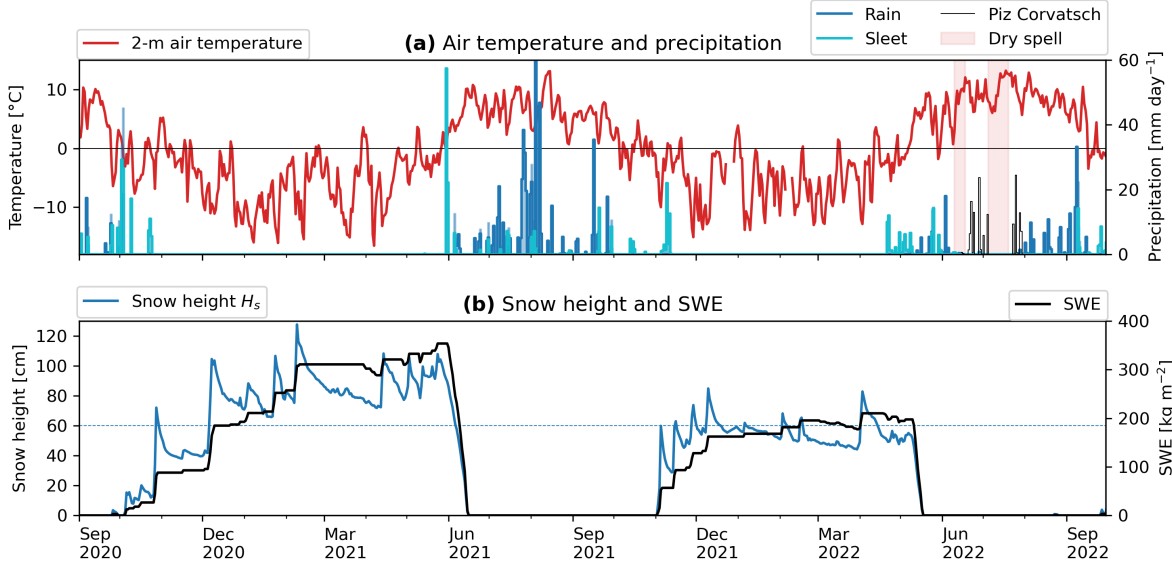

**Figure 3.** Meteorological conditions. **(a)** Air temperature (daily mean) and precipitation (daily sum). **(b)** Snow height and SWE. Rain and sleet separated based on a wet-bulb temperature threshold of $2°C$ resulted from comparing Bulk snowmelt occurred in June 2021 with that in May 2022. Precipitation data at *Piz Corvatsch* from MeteoSuisse.

## 5.2 Snow and ground thermal–hygric conditions

During the snow-rich winter 2020–2021, a thick and insulating snow cover (exceeding $70 \text{ cm}$) sealed the cavity from the atmosphere. The cavity air was kept isothermal (within $\pm 0.2 °C$; Fig. B1; Appendix Sect. B) and isohume at saturation at a much higher moisture level than that in the (colder) atmosphere (Fig. B2; Appendix Sect. B). Virtually no dry-cold air from the 420 atmosphere was mixed into the closed cavity system. A stable winter equilibrium temperature (WEqT; BTS, sensu Haeberli (1973); Haeberli and Patzelt (1982)) was reached in March 2021 ($-4.59 \pm 0.05°C$). In contrast, during the snow-poor winter 2021–2022, the rapidly fluctuating temperature and humidity (relative and specific) indicated a connection across the thin snow cover and an exchange with the dry cold air from the atmosphere. Following snow melt-out (end of zero curtain) and re-connection with the atmosphere, the cavity air began to warm and 'desaturate' from the surface downwards (July 2021; June 425 2022).

In summer, the near-surface cavity roof was generally warmer and more humid compared with both the atmosphere (despite slightly lower relative humidity) and the deeper cavity (specific humidity profiles are shown in Appendix Fig. B3). Daily average specific humidity in the cavity roof and that in the atmosphere were strongly correlated ($R^2 = 0.868$). The near-surface





cavity consistently presented a moisture surplus in relation to that of the atmosphere throughout the year. This surplus persisted

even during dry spells. Summertime in-cavity temperature gradients were the more stable the higher the surface temperature was. In summer 2021, frequent passages of cold fronts with rapid atmospheric cooling destabilised the in-cavity air column by reducing the temperature gradients. In summer 2022, dry spells impacted the sub-surface moisture conditions down to $2\,\mathrm{m}$ within $3-6\,\mathrm{days}$ after the last precipitation event: water infiltrated within minutes, near-surface relative humidity started to decrease within hours and the cavity air drying front (evaporation front at the isohume of $\mathrm{rH} = 100\%$) receded to greater

depths within days. At a depth of $2\,\mathrm{m}$, saturation was lost $\sim 5\,\mathrm{days}$ after the last precipitation event. Consequently, the in-cavity humidity gradients reversed, indicating a switch from downwards to upwards humidity transport. This was most pronounced in July 2022, during the intense mid-July dry spell accompanying a heat wave (Appendix Figs. B3, B2). In contrast, near the surface, the measured gradient in specific humidity between the near-surface AL and the atmosphere and, therefore, $Q_{LE}$ remained largely constant regardless of the different weather conditions and the humidity gradient within the AL (Sect. 6.3.3).

We determine the snow height necessary to close the snow cover and to decouple the AL from the atmosphere with the sub-surface AL air temperature, specific humidity (not shown, correlated with air temperature) and airflow speeds (Fig. 4). With increasing snow depth, the differences in air temperature and specific humidity between the cavity and the atmosphere increased (Appendix Figs. B1, B2), the correlation between in-cavity and atmospheric signal was lost and rapid (hourly–daily) fluctuations in the in-cavity temperature and specific humidity weakened (Fig. 4a). The decoupling proceeded gradually with

snow depth (sketched by the schematic envelope), the thresholds are only approximate values. Snow depths between 20 and $40\,\mathrm{cm}$ rarely occurred in the studied period, the gap (marked by '??') is a time period bias rather than an actual decoupling. Also, the sub-surface airflow speed was attenuated gradually according to the micro-topographic setting that controls the local snow depth (Fig. 4b): on gently sloping and less rough terrain, the vertical connection between the coarse-blocky AL and the atmosphere was reduced at snow heights of 5–20 cm and lost at $\sim 50-60\,\mathrm{cm}$ of snow (WS/3 and WS/4); on wind-swept ridges

with wind erosion, a much thicker snow cover was necessary to shut down the last snow funnel (Bernhard et al., 1998; Keller and Gubler, 1993) ($\sim 80\,\mathrm{cm}$ at WS/5 at the up-wind side of a big block on a ridge).

## 5.3 Surface energy balance

Monthly SEB (Eq. 5) is dominated by the short- $S^*$ and long-wave $L^*$ radiation components, followed by the sensible $Q_H$ and latent $Q_{LE}$ turbulent heat fluxes and the ground heat flux $Q_G$ (Fig. 5; Table 2). In winter, turbulent fluxes compensate

for the energy lost by net radiation. In summer, turbulent fluxes export roughly 90% of the available net radiation. Snowmelt $Q_M$ absorbs practically the entire net radiation ($Q_M \approx -Q^*$) in the respective melt-out month (June 2021; May 2022), and the sensible and latent turbulent fluxes either are then small or roughly cancel each other (Fig. 6). The ground heat flux $Q_G$ is negative (heat transfer into the ground) during snow melting (infiltration of meltwater into the frozen coarse-blocky AL liberates the latent heat) and net negative during the thaw season. The sensible rain heat flux $Q_P$ is a negligible SEB component. In short,

$S^* \gtrsim L^* \gg |Q_H| > |Q_{LE}| > |Q_G| \ggg Q_P$. The parameters and constants used in the calculations are tabulated in Appendix Sect. E (Table E1).





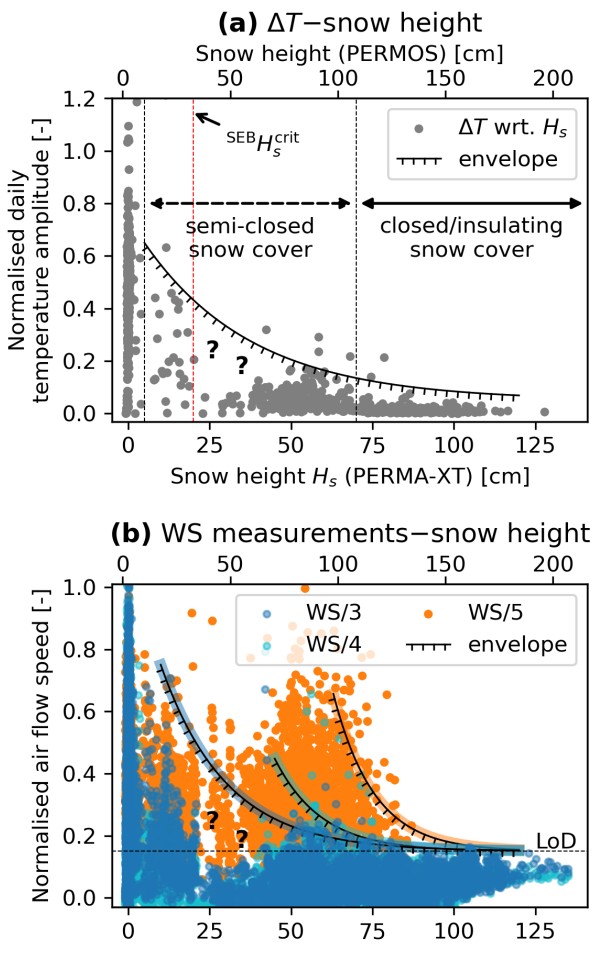

**Figure 4.** Indicators of AL–atmosphere coupling. **(a)** Relationship between the daily temperature amplitude in the cavity roof (normalised by the 2-m temperature amplitude) and the corresponding snow depth $H_s$ measured on a wind-swept rock glacier ridge (PERMA-XT station) and a broad flat area (PERMOS station). **(b)** Normalised airflow speed ($u/u_{max}$) at different locations as a proxy for sub-surface ventilation and convective coupling. Measurements below the level of detection (LoD) were considered zero. Airflow speed decreases with increasing snow height relative to the maximum speeds under snow-free conditions. Onset of decoupling varies with sensor location. WS/5 is beneath a large wind-exposed block on a ridge where snow funnels remain open longer than those on flat terrain (WS/3, WS/4).

### 5.3.1 Surface radiation

Short- and long-wave radiation are by far the largest SEB components (Fig. 5a). Consequently, the well-known effect of the snow cover on albedo is the single-largest control of net radiation and hence of the entire SEB (Fig. 5b).





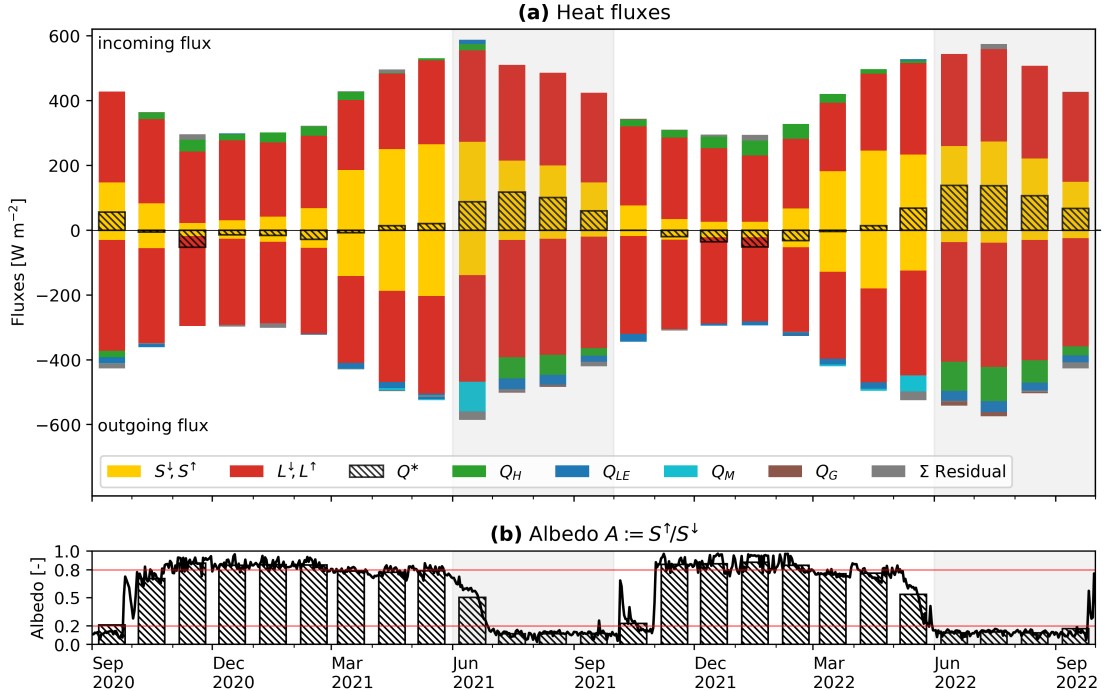

**Figure 5. (a)** Monthly energy balance components. Shown turbulent fluxes calculated using the modified Louis (1979) parametrisation ($cL'$). **(b)** Daily and monthly surface albedo $A$ as a snow-cover indicator.

In mid-winter, the meso-scale relief controls the solar radiation budget. The shaded north-facing Murtèl cirque receives no direct insolation from November to February. Depending on cloud cover and amount of incoming diffuse or terrain-reflected short-wave radiation, the net radiation is negative and dominated by the long-wave radiation budget ($L^* \ll S^* < 0$). Net radiation $Q^*$ was less negative in the cloudy and precipitation-rich winter 2020–2021 (December–February; average: $-19 \text{ W m}^{-2}$) than in the sunny and dry winter 2021–2022 ($-40 \text{ W m}^{-2}$) due to greater incoming long-wave radiation $L^{\downarrow}$ emitted by the clouds (233 vs. $215 \text{ W m}^{-2}$). A case in point is the exceptionally warm and sunny November 2020, which received in total $33 \text{ W m}^{-2}$ less short- and long-wave radiation than November 2021. The monthly mean radiation deficits are up to $-53 \text{ W m}^{-2}$, which, together with the steep snow-covered slopes, is a favourable setting for strong katabatic winds.

### 5.3.2 Turbulent heat fluxes

The Bowen ratio partitions the available energy from net radiation, the ground heat flux and the snow melt energy into sensible and latent turbulent fluxes. Throughout the seasons of both years, the heat fluxes were as follows (Fig. 6): During winter time, from November to March without direct insolation in the shaded cirque, the energy loss of $20–50 \text{ W m}^{-2}$ due to long-wave emission and terrain shading was largely compensated by the sensible heat flux $Q_H$. Latent heat flux $Q_{LE}$ at the cold snow surface was small, with mostly resublimation (moist winter 2020-2021) and sublimation fluxes (dry winter 2021-2022) within



**Table 2.** Season-averaged fluxes and Bowen ratio ($\bar{\text{Bo}} := \bar{Q}_H / \bar{Q}_{LE}$).

| Flux | Hydrological year 2020–2021 | | | Hydrological year 2021–2022 | | |
|---|---|---|---|---|---|---|
| [W m$^{-2}$] | Oct–Feb | Mar–May | Jun–Sep | Oct–Mar | Mar–May | Jun–Sep |
| $Q^*$ | $-23.4$ | $8.5$ | $91.7$ | $-27.9$ | $25.9$ | $111.9$ |
| $S^*$ | $10.9$ | $56.3$ | $155.3$ | $17.0$ | $76.2$ | $193.4$ |
| $L^*$ | $-34.3$ | $-47.8$ | $-63.2$ | $-44.9$ | $-50.4$ | $-81.5$ |
| $Q_H$ | $30.0$ | $13.4$ | $-31.3$ | $37.4$ | $18.9$ | $-71.5$ |
| $Q_{LE}$ | $-2.9$ | $-17.7$ | $-17.5$ | $-11.7$ | $-11.3$ | $-27.2$ |
| $\bar{\text{Bo}}$ | $-10.31$ | $-0.76$ | $1.79$ | $-3.21$ | $-1.67$ | $2.63$ |
| $Q_S$ | $-0.4$ | $-0.7$ | $0.0$ | $0.2$ | $-0.4$ | $0.0$ |
| $Q_G$ ($\tilde{Q}_{r_{al}}$)$^a$ | $-0.1$ | $-1.5$ | $-5.5\,(-5.2)$ | $0.5$ | $-1.2$ | $-8.7\,(-9.0)$ |
| $Q_M$ | $0.0$ | $-4.3$ | $-22.6$ | $0.0$ | $-20.4$ | $-0.1$ |
| $\Sigma$ (imbalance) | $3.3$ | $-0.8$ | $14.8$ | $-1.5$ | $11.5$ | $4.4$ |

Turbulent fluxes calculated using the modified Louis (1979) (cL$'$) bulk parametrisation. $^a$Note AL radiative flux $\tilde{Q}_{r_{al}}$ is derived from direct pyrgeometer measurements and is not within the SEB imbalance $\Sigma$.

$-6 \pm 10\ \text{W m}^{-2}$ (Fig. 6a). With the latent heat flux small and in variable direction, the computed Bowen ratio showed a large
scatter ($|\text{Bo}| > 1$; Fig. 6b). In the drier winter 2021–2022, $Q_H$ had to compensate for the heat lost by sublimation additional
to the radiative heat loss. During spring, from April to May, before the main snow melting phase, the magnitude of the fluxes
remained similar, but the direction reversed. In summer, after complete snow melt-out, from June/July to October, the turbulent
fluxes were larger to compensate the large radiation surplus. In the more rainy summer 2021, sensible and latent heat export
were similarly large ($\text{Bo} = 1.25 \pm 2.71$), whereas in the drier summer 2022, heat export was more dominated by sensible flux
($\text{Bo} = 1.48 \pm 2.47$). Towards early autumn, with decreasing temperatures, the sensible flux lost importance relative to the latent
flux (Bo decreased to approach zero). In September/October, with mixed precipitation (sleet) and first snow falls at still warm
conditions near $0°\text{C}$, the latent heat export by sublimation from the snow cover intermittently dominated over the sensible heat
uptake (Bo between $0$ and $-1$). With further cooling, the moisture supply became limited again, and sensible heat uptake took
over to offset the increasing radiation deficit ($\text{Bo} < -1$).

The measured eddy-covariance flux (buoyancy flux) falls short of closing the SEB (Fig. 6), roughly by 50–75% depending
on how large the (unmeasured) eddy latent flux is. The imbalance ('missing' flux to close the SEB (Foken, 2008)) is, in any
case, substantial.

     We compare the estimates of turbulent fluxes (monthly averages) among each other and with the net radiation (Fig. 6). In
the snow-free summer months, the Louis (1979) bulk fluxes tend to slightly underestimate, while the Businger–Dyer fluxes
(cB&D) clearly overestimate the fluxes.



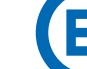 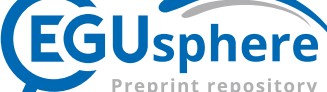

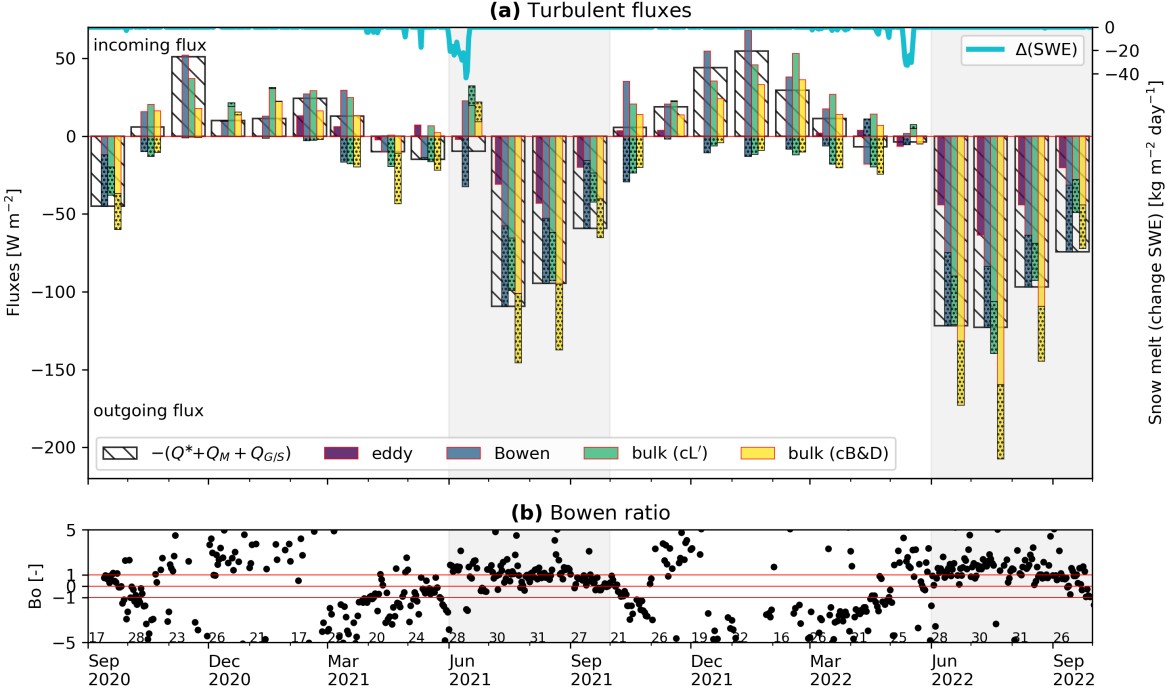

**Figure 6. (a)** Sensible and latent (dotted boxes) turbulent surface fluxes. Available energy from net radiation $Q^*$, snowmelt $Q_M$ and ground heat flux $Q_G$ (hatched boxes) is partitioned into sensible $Q_{H^{Bo}}$ and latent $Q_{LE^{Bo}}$ turbulent fluxes according to the Bowen ratio. Bowen SEB is closed by design. Bulk fluxes differ according to the stability function and do not necessarily close the SEB. **(b)** Bowen ratio (Eq. 7). November–March: large scatter in winter because $Q_{LE^{Bo}}$ is small and changes direction (unstable calculation). June/July–October: Bo $\sim$ 0.5–2.5, reflecting the wet-cool summer 2021 and the dry-hot summer 2022 (Fig. 3). Monthly average based on 16–31 valid values per month.

### 5.3.3 Snow melt energy

The energy absorbed by the melting snowpack $Q_M$ largely consumes the available net radiation in the respective melt-out months and is roughly as large as typical summer sensible heat fluxes (Fig. 5).

### 5.3.4 Precipitation heat flux

The sensible rain heat flux $Q_P^{rain}$ exerts a negligible cooling effect of $\leq -2\,\mathrm{W\,m^{-2}}$ at daily time scale (not shown). Short but intense rainfall (thunderstorms) with fluxes up to $-100\,\mathrm{W\,m^{-2}}$ in 10 min are averaged out because such high-precipitation events are short. The heat fluxes $Q_P^{solid}$ arising from mixed precipitation or a shallow snow cover that melts within hours ('summer snow'; $10-30\,\mathrm{W\,m^{-2}}$ daily average) are similar to $Q_G$ and not negligible. Such events typically occur in early



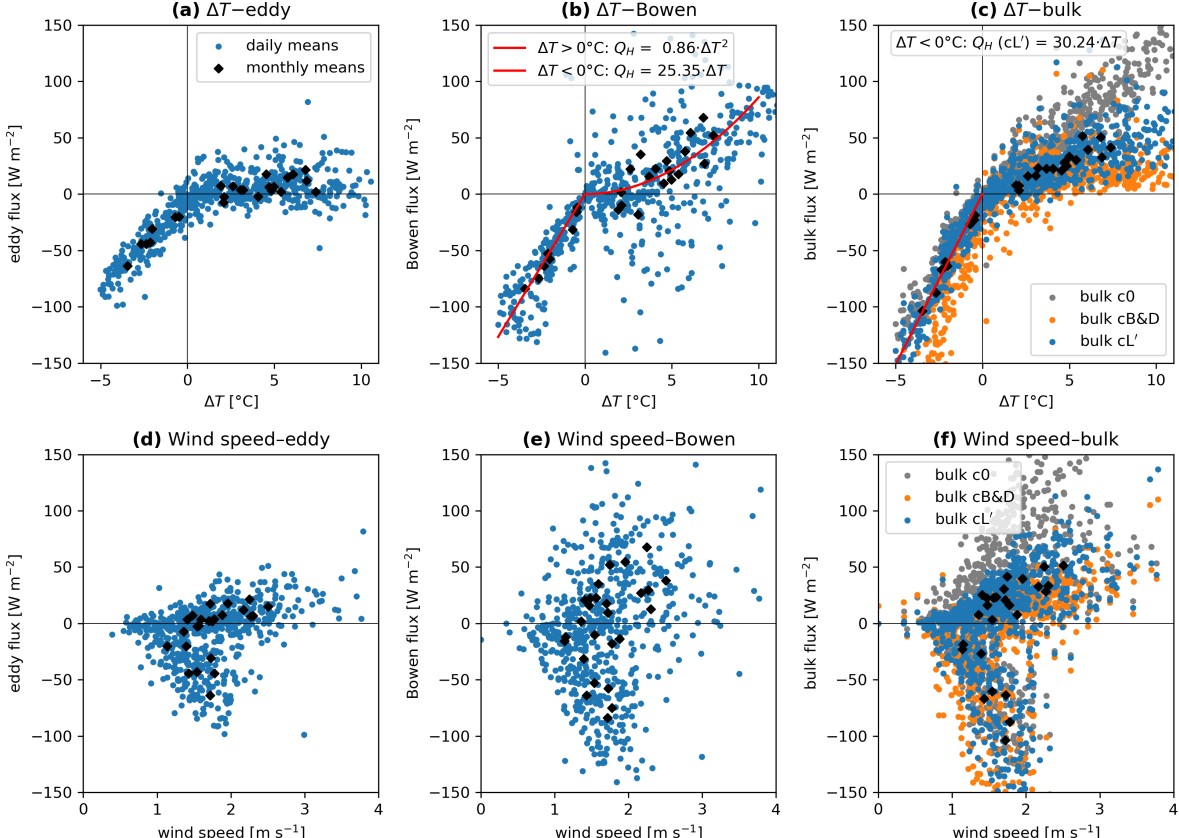

**Figure 7.** Comparison of the turbulent sensible flux estimates with the driving temperature difference $\Delta T$ (**a–c**) and measured wind speed $u$ (**d–f**) motivated by Eq. 11 (daily and monthly averages). Measured winter eddy fluxes weakly sensitive to local $\Delta T$ (**a**) and $u$ (**d**). Sensible Bowen flux shows a seasonally differing $Q_H$–$\Delta T$ relation: quasi-linear in summer and quadratic in winter (**b**) (with best-fit lines). Relation to wind speed $u$ also differs seasonally (**e**), with a clearer increase of $Q_H$ with wind speed in summer. Note that wind speed is not an input variable for the Bowen parametrisation. The seasonally differing $Q_H$–$\Delta T$ relation reflect the seasonally differing atmospheric stability and wind conditions: stronger dependency on $\Delta T$ in the unstable summer atmosphere. The quadratic relation found for the wintertime fluxes suggest that katabatic nocturnal drainage flows govern the turbulent heat transfer (Oerlemans and Grisogono, 2002). Bulk fluxes (**c**) show different sensitivities to $\Delta T$ by design: 'c0' and 'cB&D' are over-sensitive to $\Delta T$ in winter and summer, respectively.

autumn (September–October), but also occurred throughout the wet-cool summer 2021 (June–October). Their timing often
coincides with episodes of rapid ground cooling (Fig. 8).

### 5.3.5   Snow and ground heat fluxes

Snow heat flux $Q_S$ is in the range of 2 to $-4$ W m$^{-2}$, and ground heat flux $Q_G$, in the range of 30 to $-20$ W m$^{-2}$ at daily time scale (Fig. 8). The snowpack is a thermal insulator. Within-snowpack conductive fluxes are 2–3 orders of magnitude smaller



than typical SEB components. The conductive snow heat flux is negligible compared with the overall SEB and across-snowpack
convective fluxes (snow funnels; Fig. 8; 4). The simplifications from the single-layer snowpack that ignores vertically varying
snow density do not detract from this finding.

Heat storage changes in the snowpack (Fig. 8a) and uppermost coarse-blocky AL (Fig. 8b) are asymmetric (Guodong et al.,
2007) in opposite directions. In the snowpack, downwards heat transfer or warming (storage gain) is more intense (larger
maxima) but less frequent, likely due to non-conductive fluxes from refreezing meltwater (synchronised with warm spells
$T_a > 0°C$ or rapid warming to the zero curtain). In the near-surface AL, upwards heat transfer or cooling (storage loss) is
more intense but less frequent. Different weather conditions in the two summers analysed are reflected by $Q_G$: the passage of
cold fronts in summer 2021 led to more frequent convective cooling, often accompanied and enhanced by sleet; conversely,
ground warming is pronounced during dry spells in summer 2022. Similarly for the two winters: virtually no storage changes
($Q_G \approx 0$) occurred in the snow-rich winter 2020–2021, whereas temperature and sensible heat storage fluctuations continued in
the snow-poor winter 2021–2022, albeit strongly attenuated compared with those in the snow-free season (Fig. 4; Sect. 6.3.1).
The non-zero $Q_G$ in winter 2021–2022 beneath the thin snow cover reflects convective heat exchange across the snow funnels
since the conductive snow heat flux $Q_S$ is too small to account for $Q_G$.

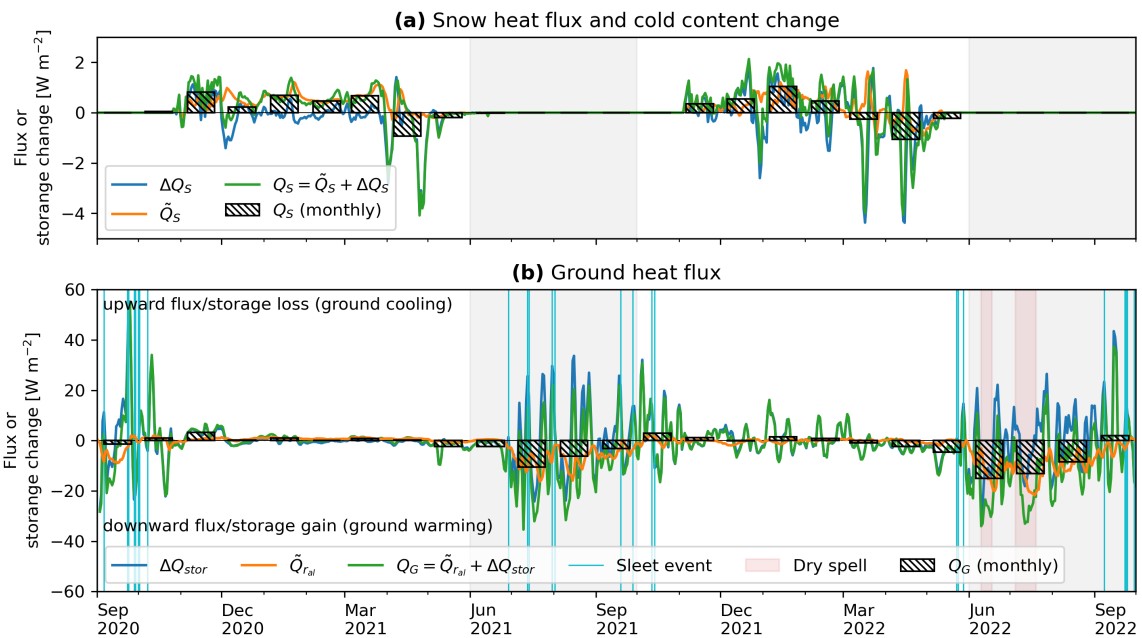

**Figure 8.** Ground $Q_G$ and snow heat flux $Q_S$. **(a)** Snow heat flux and changes in cold content of the snowpack. **(b)** Sensible storage changes
and long-wave net radiation in the instrumented cavity (uppermost $1.5$ m of the coarse-blocky AL). Beneath the insulating snow cover in
winter 2020–2021, the ground thermal regime is stable, and $Q_G \approx 0$. In contrast, the rapid fluctuations in $Q_G$ during the snow-poor winter
2021–2022 are faster than those during the conduction time, indicating convective processes as a driver. Heat input was larger in summer
2022 than in summer 2021, with cumulative positive storage changes in both summers.



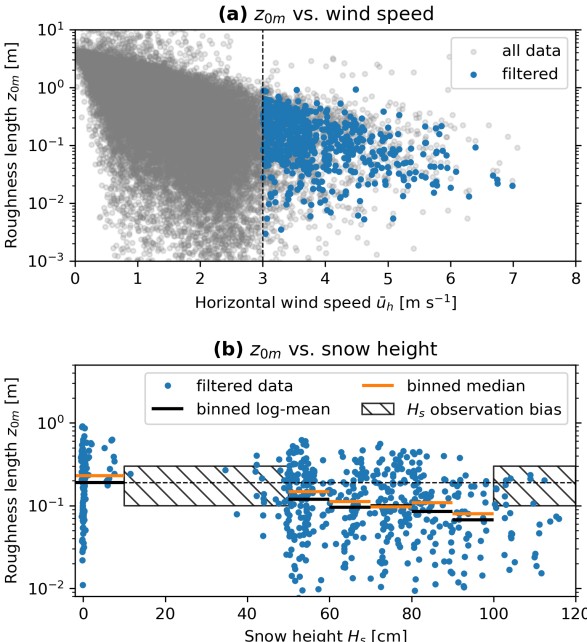

**Figure 9.** Calculated momentum roughness length $z_{0m}$ vs. **(a)** horizontal wind speed $\bar{u}_h$ and **(b)** snow height $H_s$. Of all values (grey dots), 2.1% met the quality criteria (blue dots). Momentum roughness length decreases from 0.19 to 0.07 m ($\times 10^{-0.43}$) with snow height increasing from 0 to 100 cm.

## 5.4 Aerodynamic roughness length

Valid values of roughness length for momentum $z_{0m}$ (Eq. 13) scatter over two orders of magnitude in the range of $10^{-2}$ to 525 $10^0$ m. Few (2.1%) data points met the quality criteria $u_* > 0.1\,\mathrm{m\,s^{-1}}$, near-neutral conditions $|\zeta| < 0.05$ or $\bar{u} > 3.0\,\mathrm{m\,s^{-1}}$ (Radić et al., 2017; Nicholson and Stiperski, 2020) (Fig. 9a). Bin-wise average (median) is 0.19 m (0.23 m) for snow-free conditions ($H_s < 10$ cm) and slightly decreases with increasing snow height up to 0.07 m (0.08 m) at 90–100 cm of snow (Fig. 9b). Averages were calculated from the average of the logarithmised values (Radić et al., 2017). Snow heights between 10 and 50 cm or exceeding 100 cm are rarely observed in the two-year data set, hence the observation gap. $z_{0m}$ is itself a 530 function of sensor distance and hence of snow height $H_s$ (Eq. 13). To control for possible spurious correlation in the $z_{0m}$–$H_s$ relation, we eliminated the confounding variable $H_s$ and tested with the constant measurement height $z_{\mathrm{CSAT}} = 3.78$ m. This did not significantly affect the $z_{0m}$–$H_s$ relation.



## 6 Discussion

### 6.1 Surface fluxes and uncertainties

#### 6.1.1 Fluxes

Monthly SEBs are closed within the calculation uncertainties of $\leq 20$ W m$^{-2}$ (Fig. 10 blue bars; Table 3). This represents a substantial improvement compared with Mittaz et al. (2000) and Scherler et al. (2014) and is due to the novel sensor array used in the present work.

The largest residual (sensu Scherler et al. (2014); imbalances, sensu Mauder et al. (2020)) occur during the transition between seasons, when ground thermal conditions linger around $0°$C, during extreme meteorological conditions or in mid-winter (December–March). These are the snow melt-out months (June 2021, May 2022), early autumn (September), and the July 2022 heat wave and accompanying dry spell that strongly impacted the ground thermal and moisture regime (Figs. B1–B2). Larger deviations that occurred in the snow-rich mid-winter 2020–2021 are reduced by a 'katabatic wind correction' for the variable anemometer height above the snow surface (Sect. 6.2).

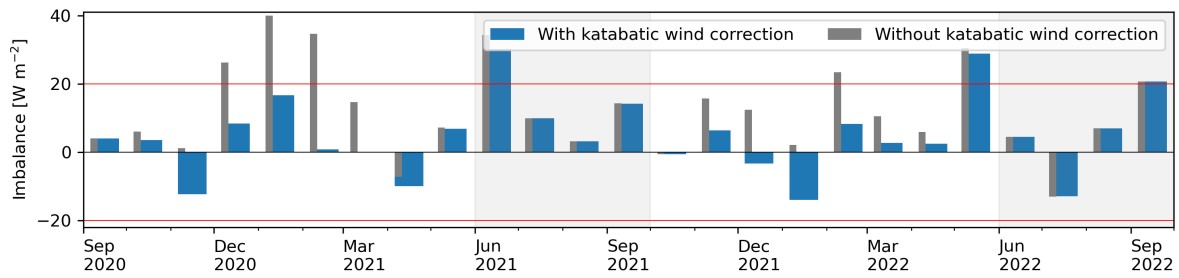

**Figure 10.** SEB imbalance (monthly averages). Turbulent fluxes calculated using modified Louis (1979) (cL$'$) bulk parametrisation. Correcting excessively high wind speed measured in the low-level katabatic jet at thick snow cover improves the SEB in the snow-rich winter 2020–2021 (katabatic wind correction; Sect. 6.2, Eq. 23).

#### 6.1.2 Parameter sensitivity

Uncertainties arise from terrain or snow cover variability across the rock glacier, spatially variable properties of the coarse-blocky AL (e.g. emissivity, porosity, intrinsic permeability) and instrumental measurement errors, among other factors. We assessed the impact of the largest sources of parameter uncertainty on the heat fluxes (Table 3; sensor accuracy from Scherler et al. (2014); Hoelzle et al. (2022)). We estimate our overall accuracy as $\sim 20$ W m$^{-2}$ at daily time scale, due to uncertainties in $Q_G$ (uncertainties in sensible storage changes $\Delta Q_{stor}$) and $Q_H$ (uncertainty in surface temperature $T_s$ propagated from the emissivity $\varepsilon$; Eq. 9). These uncertainties play an important role in this study. The SEB uncertainties determine which processes are included in the SEB estimates and which not, complementing measurement-driven criteria. We consider insignificant (i.e.



within the uncertainty) the processes with associated fluxes smaller than our $20\,\mathrm{W\,m^{-2}}$ uncertainty threshold. The processes considered insignificant in the context of the SEB are not necessarily insignificant at depth. In fact, in the AL, all daily-averaged

sub-surface heat fluxes are within $20\,\mathrm{W\,m^{-2}}$ (Fig. 8b). We apply this criteria on the nocturnal Balch ventilation (Sect. 6.3.2) and the decoupling snow height $^{\mathrm{SEB}}H_s^{\mathrm{crit}}$ (Sect. 6.3.3).

Insolation differences due to the local shading effect of the coarse terrain surface, the terrain slope effects or instrumental effects (cosine response) might lead to differences up to $30\,\mathrm{W\,m^{-2}}$ (daily average) in summer (Table 3) (Otto et al., 2012; Scherler et al., 2014; Fitzpatrick et al., 2017; Kraaijenbrink et al., 2018). Uncertainties in the outgoing long-wave radiation

$L^\uparrow$ of $\pm 10\%$ corresponding to a radiometric surface temperature difference of $7^\circ\mathrm{C}$ might arise from patchy snow cover. The lateral advective heat transport altering the boundary layer characteristics (Essery et al., 2006; Mott et al., 2013, 2015) could explain the large SEB imbalance (Fig. 5) of roughly $100-150\,\mathrm{W\,m^{-2}}$ during the melt-out phase. Smaller but still significant $T_s$ differences of $2^\circ\mathrm{C}$ can occur due to local shading (micro-topography) or uncertainty in emissivity (e.g. variable snow emissivity; Steiner et al. (2018)) and lead to considerable $Q_H$ uncertainties of up to $50\,\mathrm{W\,m^{-2}}$. Surface temperature

differences from different emissivity values ($\varepsilon = 1$ instead of Scherler et al. (2014)'s 0.96; Eq. 9) are within $-1.2$ and $+0.25^\circ\mathrm{C}$. The deviations tend to be largest on hot clear-sky days with little incoming long-wave radiation $L^\downarrow$ (Eq. 9), and hence might translate into considerable uncertainties in $Q_H$ precisely when these are largest.

Temperature and specific humidity uncertainties driving bulk fluxes (Eqs. 7; 11–12) lead to considerable $Q_H$ and $Q_{LE}$ uncertainties. The Essery and Etchevers (2004) parametrisation is moderately sensitive to the spatially variable wind field

which might arise from the micro-topography, for example wind sheltering in the furrows. The momentum roughness length $z_{0m}$ is varied by a factor of $10^{0.5}$ (between $0.09$ and $0.45\,\mathrm{m}$), which reflects the standard deviation of the measurements (Fig. 9). The roughness lengths are perhaps among the most critical parameters for the estimation of turbulent fluxes when using the bulk approach (besides the emissivity $\varepsilon$): sensitive yet hard to constrain on rough and complex mountainous terrain (Steiner et al., 2018; Rounce et al., 2015; Giese et al., 2020). Equivalence between momentum $z_{0m}$ and scalar roughness lengths for heat

$z_{0h}$ or moisture $z_{0q}$ would lead to prohibitively large deviations and can be excluded. As the rough terrain turns sensor height above surface into a somewhat vague parameter, we vary it by a typical block edge length of $0.5\,\mathrm{m}$. The arising $Q_H$ deviations of $10-20\,\mathrm{W\,m^{-2}}$ are similar to other parameter uncertainties. Air temperature and specific humidity measurements from the PERMOS and PERMA-XT stations, located within $50\,\mathrm{m}$, differ by $0.6 \pm 1.0^\circ\mathrm{C}$ and $0.004 \pm 0.24\,\mathrm{g\,kg^{-1}}$, respectively. The calculated fluxes are similar; $Q_H$ is within $\pm 5\,\mathrm{W\,m^{-2}}$, and $Q_{LE}$ within $2\,\mathrm{W\,m^{-2}}$. Maximum deviations are temporarily up

to $10-15\,\mathrm{W\,m^{-2}}$ in winter and during the snow-melt period. The PERMOS and PERMA-XT station data yield flux estimates indistinguishable within their uncertainty.

Ground heat flux $Q_G$ primarily reflects the rate of temperature change (RTC) of the near-surface AL and is weakly sensitive to the pyrgeometer flux measurements $\tilde{Q}_{r_{al}}$. The reason behind is that $Q_G$ (Eq. 20) is dominated by the sensible storage changes $\Delta Q_{stor}$ of the $1.5\,\mathrm{m}$ thick blocky layer above the long-wave radiation measurement (Liebethal et al., 2005). Consequently,

rather large uncertainties in $Q_G$ ($20\,\mathrm{W\,m^{-2}}$) come from uncertainties in the storage changes $\delta(\Delta Q_{stor})$ from two factors, namely the porosity $\phi_d$ and the thermal adjustment time $\Delta t_r$ (time lags). First, a high porosity limits the heat storage capacity that scales with $(1-\phi_d)$. We tested a plausible range of $\delta\phi_d \pm 0.2$ and obtained uncertainties up to $20\,\mathrm{W\,m^{-2}}$. Porosity might





laterally vary the most close to the surface, precisely where daily temperature amplitudes are largest. Second, the assumption of similar air and rock temperature profiles $\bar{T}_a(z) \approx T_r(z)$ – local thermal equilibrium (LTE) assumption (Nield and Bejan, 2017))

– becomes problematic in the roof of the ventilated cavity, where conditions are highly transient (convective heat transfer and effect of insolation). Not the entire rock mass might adjust to the rapid temperature fluctuations of $+2$ to $-3°$C from day to day. The chosen thermal relaxation time $\Delta t_r$ of 1 day is a minimum duration (Appendix Sect. C). We assess the influence of longer adjustment times $\Delta t_r$ by comparing 1- with 5-day running averages. The differences are similar to the uncertainties related to porosity. We conclude that, due to the large and rapid heat turnover in the ventilated near-surface coarse-blocky AL

under highly transient conditions, the ground heat flux $Q_G$ is arguably the least constrained flux. This might not be a surprising finding on a landform that does not present a clearly defined surface.

Finally, we let $^{\mathrm{SEB}}H_s^{\mathrm{crit}} = 80$ cm to simulate an open snow cover in early–mid winter 2020–2021 (October–February; Fig. 3) and throughout the snow-poor 2021–2022. Due to the large humidity differences across the snow cover (Fig. B2), this resulted in a massive (up to $10^2$ fold) increase in $Q_{LE}$. The closing of the snow cover is a key control factor for the SEB and ground

thermal regimes (Eq. 10; Sect. 6.3.1) (Hanson and Hoelzle, 2004).

**Table 3.** Instrumental sensitivity: uncertainty due to parameter values and meteorological variables.

| Parameter, variable | $\delta Q^*$ [W m$^{-2}$] | $\delta Q_H$ [W m$^{-2}$] | $\delta Q_{LE}$ [W m$^{-2}$] | $\delta Q_G$ [W m$^{-2}$] |
|---|---|---|---|---|
| $S^* \pm 10\%$ (daily total) | $\leq 30$ | | | |
| $L^\uparrow \pm 10\%$ ($L^\uparrow \propto \sqrt[4]{T_s}$) | $25-40$ | $50-150$ | | 0.2 |
| $T_s \pm 2°$C ($L^\uparrow \pm 3\%$) | $7-12$ | $10-50$ | $\leq 5$ | |
| $\Delta T \pm 1.0°$C | | $5-25$ | | — |
| $\Delta q \pm 0.5$ g kg$^{-1}$ | | | $5-25$ | — |
| $u \pm 10\%$ ($\leq 0.3$ m s$^{-1}$) | | 10 | 5 | |
| $\log_{10}\{z_{0m}\} \pm 50\%$ | | $30-80$ | $5-20$ | |
| $(z_{0h}/z_{0m}) \pm 20\%$ | | $\leq 6$ | $\leq 2$ | |
| $z_{0h},\ z_{0q} = z_{0m}$ | | $\leq 300$ | $\leq 100$ | |
| $z_u,\ z_T \pm 0.5$ m | | $10-20$ | $\leq 10$ | |
| $T_a,\ q_a$ PERMOS/PERMA-XT | | 5 | 2 | |
| $\tilde{Q}_{r_{al}} \pm 15\%$ | | | | 2 |
| $\phi_d = 0.5 \pm 0.2$ | | | | $\leq 20$ |
| $\Delta t_r$ 1–5 days | | | | $\leq 20$ |
| $^{\mathrm{SEB}}H_s^{\mathrm{crit}} = 80$ cm | | | $30-60$ | |

Uncertainty of daily average fluxes. $Q_H$ and $Q_{LE}$ using the modified Louis (1979) parametrisation (cL$'$).

In the sensitivity analysis (Table 3), input parameters and meteorological variables are varied independently from each other and based on likely maximum measurement errors. However, the meso-scale relief and local sub-surface processes



like ventilation might add some systematic bias that exceed instrumental errors. We'll explore parametrisation uncertainties (stability corrections) in Sect. 6.2 and uncertainties in the meteorological input variables in Sect. 6.3.

## 6.2 Meso-scale terrain effects on turbulent flux parametrisations

### 6.2.1 Katabatic wind

The interaction of the steep snow-covered wintertime terrain with a negative radiation balance induces down-slope katabatic winds that govern the near-surface wind field. This, in turn, affects the calculation of the bulk turbulent fluxes that require a representative wind speed as input. Our initially calculated wintertime turbulent fluxes were 'too large' by $10-35$ W m$^{-2}$ on monthly average (Fig. 10), especially during the overcast snow-rich winter 2020–2021, while the radiative cooling and the forcing temperature deficit $\Delta T$ were weaker than in the snow-poor winter 2021–2022 (Fig. B1; Appendix Sect. B; Eq. 11–12). This result suggests that measured wind speeds were off in relation to the snow height and not to the temperature deficit. In fact, wind tower measurements on Murtèl performed by Stocker-Mittaz (2002) showed strong and persistent katabatic winds in winter, with a wind speed maximum a few metres above the surface weakly correlated to snow height (Appendix Sect. D; cf. Fig. 2). The growing snow cover 'shifted' the high-velocity region of the low-level katabatic jet to the level of the wind sensor, causing an apparent wind speed increase. We compensate the wind speed $u_s$ measured at variable height above the snow cover $(z_s - H_s)$ for the snow height with a simple power-law relation:

$$u = u_s(z_u/(z_s - H_s))^{-\hat{m}}, \tag{23}$$

where $\hat{m}$ is the exponent. Note that the usual correction of the sensor height above the variable snow surface further increased rather than decreased the turbulent fluxes. Since our measurements cannot resolve the near-surface wind speed profile of the low-level katabatic jet, the intention of this ad hoc 'katabatic wind correction' is not to accurately describe the wintertime wind profile but rather to pragmatically render our input data amenable to the flux-gradient parametrisations based on the Monin–Obukhov theory. We use $\hat{m}$ as a calibration parameter (denoted by the circumflex). A literature value of $\hat{m} = 0.6$ for stable conditions (Arya, 2001) reduced the wind speed sufficiently to yield turbulent fluxes that close the SEB within our uncertainty threshold (Fig. 10). Although the Monin–Obukhov theory is not strictly valid under such conditions (Grisogono et al., 2007), our consistent flux estimates based on a scaled wind speed are in line with Endrizzi et al. (2014); Denby and Greuell (2000), who argue that the bulk method still provides reasonable flux estimates when measured close to the surface, as was the case here (measurements within 2 m above the variable snow cover). This illustrates the importance of accurate wind speeds for the calculation of turbulent fluxes. We suggest an alternative solution in Sect. 6.2.2.

### 6.2.2 Comparison of turbulent flux parametrisations

From the comparison of the measured eddy-covariance fluxes and the calculated Bowen and bulk fluxes (daily averages), we reach the following conclusions:



Overall, the modified Louis (1979) parametrisation (cL′) seems the best choice to parametrise the turbulent fluxes on Murtèl for three reasons: First, as expected, the modified Louis (1979) 'cL' bulk fluxes resulted near-identical to the iteratively calculated Monin–Obukhov fluxes (Fig. 11d; $r = 0.996$), but at less computational cost. Second, no filtering/post-processing was necessary because virtually all estimates met the quality criteria. Third, these show the least deviations to the Bowen fluxes (Fig. 11e), which we consider the benchmark as the Bowen SEB is closed by design. The Bowen fluxes offer reasonable daily estimates based on minimal measurement requirements, namely temperature and specific humidity profiles. Wind speed is not required. The minimum time resolution is daily scale. Hourly values are either numerically unstable (small $\Delta T/\Delta q$, especially over a 'warm' spring–early-summer snow cover) or do not account for the systematic diurnal wind speed variations. In summer, for example, hourly sensible Bowen fluxes are overestimated during the calm nights and compensated by excessive daytime fluxes.

The eddy-covariance flux systematically underestimates the sensible turbulent flux by a factor of $\sim 2$, leading to a large imbalance (underclosure) in the eddy SEB. The error from the lacking SND-correction (Sect. 3.2.2) cannot account for the eddy-SEB imbalance. The ratio between the eddy sensible flux and the (sonic) buoyancy flux is between 0.93 and 1.05 (10 and 90% quantile, respectively; Eq. 2) on a daily average. Since both the air and the ground surface are most often far from saturation, the eddy sensible flux error due to the lacking high-frequency gas analyser is less than $10\%$ on 617 out of 692 days (89.2%) with valid Bowen ratios. Larger deviations occur on the remaining 75 days with $|Q_{LE}| > |Q_H|$ (Bowen ratio within $-0.6$ and 0.5), typically during snow melting ('warm' saturated snow surface), cloudy-rainy summer days (more frequently in summer 2021 than 2022) or during the first snowfall in autumn. Applying the SND correction showed little added value. Nonetheless, the summertime eddy-covariance fluxes correlate reasonably well with the Bowen sensible turbulent fluxes (Fig. 11a) and the modified Louis (1979) bulk fluxes (Fig. 11b) or the driving temperature gradient (Fig. 7a), and therefore random instrumental or data processing errors represent an unlikely explanation. We hypothesise that some hidden systematic reason causes the flux underestimation and the large eddy-SEB imbalance, e.g., secondary circulation (Foken, 2008) of the anabatic afternoon winds.

The empirical Businger–Dyer ('cB&D') formulation, developed over flat terrain, overestimates the turbulent fluxes and deviates more strongly for larger negative fluxes ('banana' shape; Fig. 11f), despite extensive filtering. Even the bulk parametrisation without stability correction ('c0') outperformed 'cB&D' for the summer fluxes under unstable atmospheric conditions (Fig. 11c). This finding agrees well with Steiner et al. (2018) on the *Lirung* debris-covered glacier that shares many topoclimatic features with the Murtèl rock glacier (unstable atmosphere over a strongly heated debris surface with anabatic valley winds).

Finally, we describe the relation between the wintertime sensible Bowen fluxes with the driving temperature gradient $\Delta T$ ('surface temperature deficit') as quadratic (Fig. 7b). A quadratic functional $Q_H$–$\Delta T$ relation is unique to Oerlemans and Grisogono (2002)'s katabatic model. Note that the Bowen fluxes are independent of wind speed (Eq. 7). Together with the ad hoc 'katabatic correction', our data shows that the turbulent fluxes in a snow-covered cirque with katabatic winds might be better parametrised by a katabatic model than by the common Monin–Obukhov bulk method, as also found by Radić et al.



(2017) on a sloped glacier surface and discussed by Grisogono et al. (2007) in the context of atmospheric boundary layer modelling.

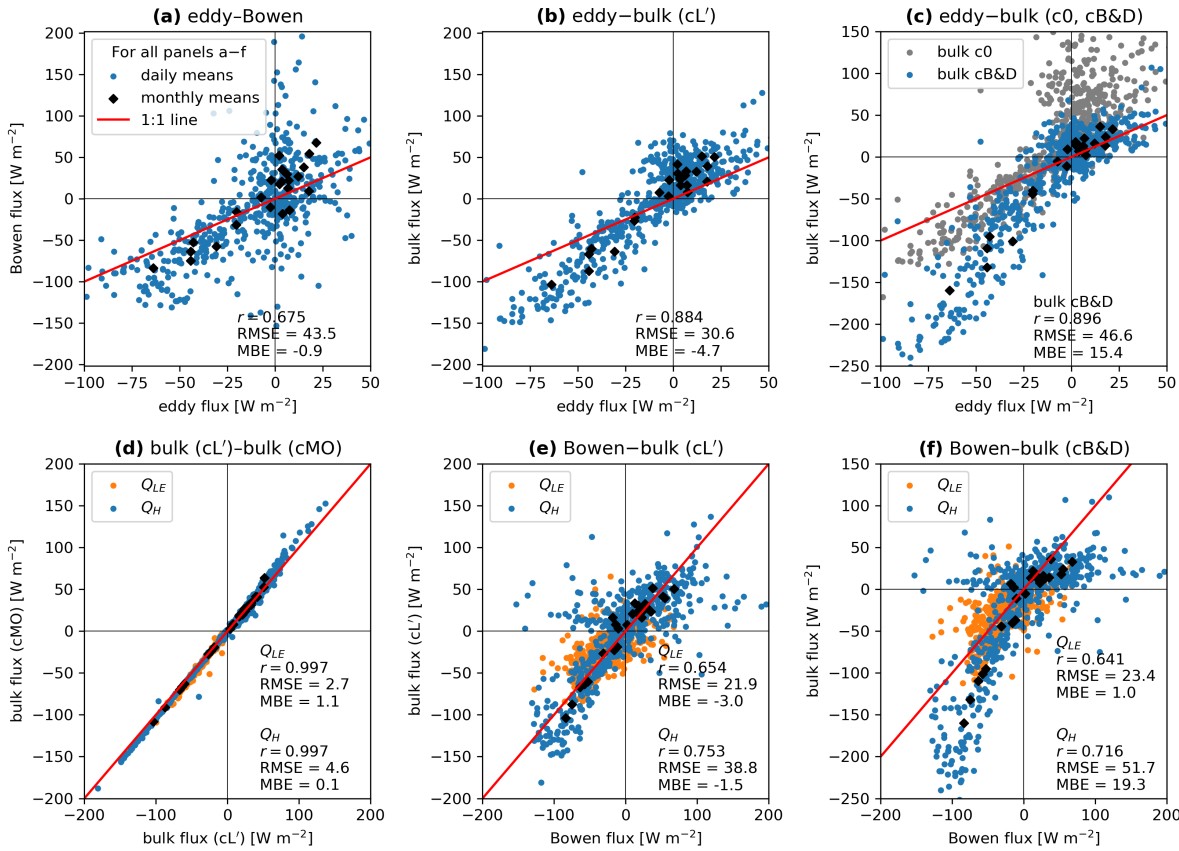

**Figure 11.** Comparison of turbulent flux estimates daily and monthly averages (cf. Fig. 6 for time series). Comparison metrics: root mean square error (RMSE [W m$^{-2}$]), mean bias error (MBE [W m$^{-2}$]) and Pearson correlation coefficient ($r$ [-]) on daily estimates.

## 6.3 Micro-scale landform effects on turbulent flux parameters

The 'surface' temperature $T_s$ and 'surface' specific humidity $q_s$ are key inputs for the Bowen ratio (Eq. 7) and bulk methods to estimate turbulent fluxes (Eqs. 11–12). However, heat and moisture can be drawn from the ventilated AL beneath the surface, provided the snow cover is sufficiently thin to allow convective exchange ($H_s < {}^{\mathrm{SEB}}H_s^{\mathrm{crit}}$). Although an 'open' snow cover that allows vertical convective exchange between the AL and the atmosphere (Sect. 6.3.1) shapes the wintertime ground thermal regime up to a snow height of $\sim 70\,\mathrm{cm}$, we will argue in Sects. 6.3.2 and 6.3.3 that already at more than $20\,\mathrm{cm}$ of snow
convective exchange across the snow cover can be ignored in the context of our SEB.



### 6.3.1 Snow cover and AL–atmosphere coupling

The snow cover controls the coupling between the coarse-blocky AL and the atmosphere. The turbulent fluxes draw heat and moisture from the AL as long as the snow cover is 'open' via snow funnels, a phenomenon widely observed on coarse-blocky landforms (Sawada et al., 2003; Delaloye et al., 2003; Delaloye and Lambiel, 2005; Morard et al., 2008; Schneider, 2014; Kellerer-Pirklbauer et al., 2015; Popescu et al., 2017). A thickening snow cover gradually suppresses the vertical convective coupling between the AL and the atmosphere, as more snow funnels close (sketched by the schematic envelopes in Fig. 4). A snow height of $\sim 10$ cm begins to decouple the coarse-blocky AL from the atmosphere ('semi-closed' in Fig. 4), but much more snow ($\sim 70$ cm) is necessary to achieve the insulating effect of the snow cover on the ground thermal regime ('closed/insulating'). Then, the snow cover is thick enough to suppress convective air exchange, rapid sub-daily fluctuations are strongly attenuated, and large temperature and moisture gradients to the outside air build up (Haberkorn et al., 2015), indicating that the AL–atmosphere coupling is weak. Such a high value is typical for a terrain as rough and blocky as on Murtèl, agreeing with e.g. Hanson and Hoelzle (2004); Herz (2006). The insulating effect of a thick snow cover was already observed decades ago and used to indirectly map the permafrost distribution using the bottom temperature of snow cover (BTS) method (Haeberli, 1973; Haeberli and Patzelt, 1982)).

As regards the SEB, a snow cover as thin as $^{\mathrm{SEB}}H_s^{\mathrm{crit}} = 20$ cm causes a decoupling that is strong enough for our SEB to be significant. This is shown in the snow-poor winter 2021–2022 with snow depths between 40 and 70 cm, when the ground heat flux $Q_G$ kept fluctuating, but remained below the 20 W m$^{-2}$ uncertainty threshold (Table 3). The convective flux across the snow cover cannot deviate strongly from $Q_G$, because it was the single-largest heat flux to supply/extract the heat for the AL sensible storage changes (small conductive snow heat flux, $Q_S \pm 2$ W m$^{-2}$, $T_s < 0°$C and no latent effects from snowmelt in that period). We take this threshold as the critical snow thickness ($^{\mathrm{SEB}}H_s^{\mathrm{crit}}$; Eq. 10). We emphasise that the 20 cm threshold is an operational definition for the 'decoupled' snow cover in the context our SEB and its uncertainties.

### 6.3.2 'Surface' temperature and near-surface ventilation

From Sect. 6.3.1 follows that for the decoupling or insulating snow cover ($H_s \geq {}^{\mathrm{SEB}}H_s^{\mathrm{crit}}$), the reference surface coincides with the snow surface. The radiometric surface temperature represents the surface temperature $T_s$ (Eqs. 7; 11). Whenever the snow cover is open or under snow-free conditions ($H_s < {}^{\mathrm{SEB}}H_s^{\mathrm{crit}}$), the radiometric surface temperature is the input meteorological variable $T_s$ (Eq. 11). The surface is where the solar radiation is intercepted and transformed to thermal energy, hence the main source of sensible heat for $Q_H$. The large gradient $(T_a - T_s)/z_T$ drives the sensible turbulent flux $Q_H$ (Eqs. 7, 11; Fig. 12a) and the wind-forced ventilation (Fig. 12b) (Panz, 2008; Stocker-Mittaz, 2002). This is shown by the measured eddy-covariance flux (Fig. 12c) that largely follows the difference between air and radiometric surface temperatures (Fig. 12a, red area), as do the local (anabatic) wind in the atmosphere above surface and the airflow speed in the near-surface AL (Fig. 12b). During daytime with strong heating on the low-albedo surface, the blocky surface exceeds air temperatures by $> 15°$C.

However, we found evidence of near-surface ventilation that cannot be parametrised by the 2-m air temperature and the radiometric ground surface temperature: Nocturnal Balch ventilation, a night-time cooling process owed to the interplay of the





air permeability and the thermal inertia (heat retention) of the coarse-blocky AL that is most apparent during fair-weather sum-
mer days with clear nights. Upwards turbulent heat export is protracted into the evening hours, long after sunset ($S^*$; Fig. 12c),
as shown by the measured in-cavity airflow speed (WS/3; Fig. 12b) and the eddy-covariance flux (Fig. 12c). Furthermore, the
measured eddy-covariance flux remains negative (upwards-directed) during the early morning hours, when the terrain surface
has radiatively cooled to air temperature $T_s \approx T_a$ (or occasionally, at clear nights, falls even below: $T_s < T_a$). The calculated
bulk fluxes driven by $2\,\mathrm{m}$ air temperature and radiometric ground surface temperature (rGST; red area in Fig. 12a; Eq. 11)
are then close to zero (or even downwards) and do not fully capture this nocturnal drawing of heat from the near-surface AL.
The large thermal inertia of the rock mass and the protection from long-wave radiative cooling stabilise the sub-surface air
temperatures and keep the air in the roof of the ventilated cavity during the nights warmer than that in the atmosphere outside
and that on the ground surface (Fig. 12a; TK1/1 $\approx$ TK6/1 $> T_a \approx T_s$ by $\sim 4°\mathrm{C}$). In the nighttime, the locally stably stratified
air in the uppermost cavity roof is hence warmer and more unstable than that in the atmosphere (non-local static stability, sensu
Stull (1991)). The high permeability of the coarse-blocky AL allows this air to escape upwards into the atmosphere, or equiva-
lently, allows the colder outside air to sink into the cavity, thereby exporting heat (*Balch ventilation* (Balch, 1900; Millar et al.,
2014)). Panz (2008) describes an occasional nocturnal sub-surface ventilation on Murtèl when $T_s < T_a$. Also Yao et al. (2014)
interpreted nocturnal near-surface ventilation on the debris-covered Koxkar glacier (Xinjiang, China) from eddy-covariance
and temperature data.

We assess this uncertainty by using the cavity roof temperature TK1/1 instead of the conventional $2\,\mathrm{m}$ air temperature as
input $T_a$ for the bulk fluxes (Eq. 11) and compare the nighttime fluxes (when TK1/1$> T_s$). Due to the low wind speeds at night,
the estimated nocturnal Balch fluxes are small despite the appreciable temperature gradients ($10-20\,\mathrm{W}\,\mathrm{m}^{-2}$), maximally
$10\,\mathrm{W}\,\mathrm{m}^{-2}$ larger than the 'conventional' bulk flux and within our SEB $20\,\mathrm{W}\,\mathrm{m}^{-2}$ uncertainty (Table 3). Since the short-
wave radiative forcing $S^*$ and the wind speed covary in phase, the 'conventional' parameters radiometric rGST and $2\,\mathrm{m}$ air
temperature are sufficiently adequate to parametrise the turbulent fluxes when sub-daily resolution is not required – especially
considering that the chosen parametrisation (stability function) shows a much larger effect on the calculated turbulent flux
(Fig. 12c; compare 'cL' with 'cB&D'). We reach a conclusion equivalent to that in Sect. 6.3.1 for the critical snow height: the
nocturnal convective processes in summer are within the uncertainties of the daily-averaged SEB but might exert an important
cooling effect on the sub-surface energy balance.

### 6.3.3 'Surface' humidity and evaporation

From Sect. 6.3.1 follows that for the decoupling or insulating snow cover ($H_s \geq {}^{\mathrm{SEB}}H_s^{\mathrm{crit}}$), the snow surface humidity at
saturation $q_{ss}^*$ represents the surface specific humidity $q_s$ (Eqs. 7; 12). Whenever the snow cover is open or under snow-free
conditions ($H_s < {}^{\mathrm{SEB}}H_s^{\mathrm{crit}}$), the humidity measurement in the cavity roof $q_{sa}$ is the input meteorological variable $q_s$ (Eq. 10).

Perhaps contrary to the impression of a dry-looking blocky surface, the estimated summertime latent turbulent flux $Q_{LE}$
is at $\sim 30\,\mathrm{W}\,\mathrm{m}^{-2}$, corresponding to an evaporation rate of $\sim 1\,\mathrm{mm}$ w.e. $\mathrm{day}^{-1}$, even during the severe July 2022 dry spell.
This value is in the range of evaporation rates reported for debris-covered glaciers of $0.6-2.8\,\mathrm{mm}$ w.e. $\mathrm{day}^{-1}$ (Collier et al.,
2014; Yao et al., 2014; Steiner et al., 2018). However, note that our evaporative flux estimate $Q_{LE}$ might be an upper bound, in





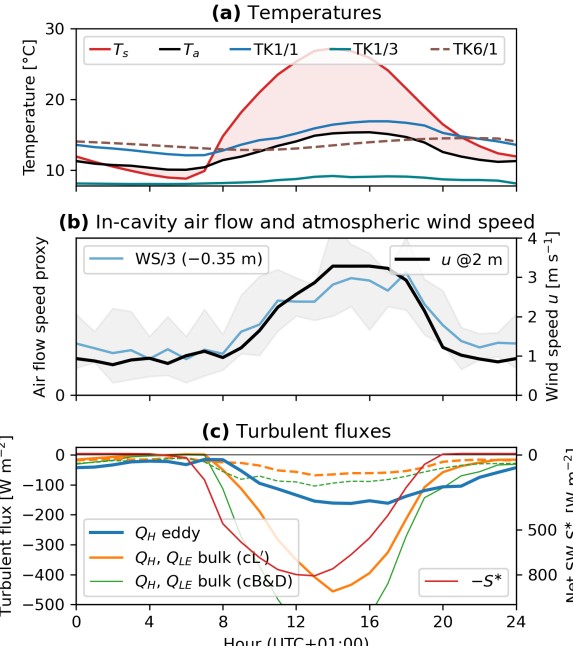

**Figure 12.** Hourly averages of **(a)** temperatures, **(b)** airflow and wind speeds, **(c)** sensible turbulent fluxes and net short-wave radiation $S^*$ during a summer fair-weather period (15–19 July 2022). The cavity air is stably stratified (TK1/1>TK1/3), but the air in the cavity roof is warmer and unstable compared with the outside air. During clear summer nights, the atmospheric and the blocky surface air cool down more strongly than the air in a near-surface coarse-blocky AL that receives heat from the blocks (air temperature TK1/1 approaches the rock temperature TK6/1 in the night). Conversely to the calculated bulk fluxes, the measured eddy flux decays slowly in the evening–midnight (18–24 h) and remains negative (upwards) despite the small temperature gradient in the early morning (3–7 h, **(a)**). This clearly shows how the turbulent flux draws heat from the shallow air layer, is nocturnally warmer than $T_a$ and thus unstable. **(b)** Atmospheric wind and airflow speeds in the instrumented cavity covary in phase (but at $\sim 10\times$ smaller magnitudes), follow the net radiation with some delay and are closely related to the turbulent fluxes **(c)**. The specific humidity shows small diurnal oscillations; the associated $Q_{LE}$ vary less than $Q_H$ in absolute terms.

particular during the July 2022 dry spell. The moisture source for evaporative $Q_{LE}$ is in the AL, not at the surface (except during precipitation events). However, our bulk parametrisation ($C_{bt}$; Eq. 12) ignores the additional resistance to vapour transport

imposed by the blocky layer between the specific humidity measurement in the cavity roof and the atmosphere ($q_{sa}$ is measured 0.7 m beneath the terrain surface; Fig. 2d). The neglected resistance to vapour transport in the coarse-blocky AL during moisture-limited evaporation stages, when moisture is drawn from deeper levels and governed by vapour diffusion through the porous AL (Pérez, 1998; Collier et al., 2014), might lead to an overestimation of $Q_{LE}$. The longer the dry spell lasts, the deeper in the AL the moisture is drawn from (Sect. 5.2), and the more important this effect of vapour transport resistance becomes.

This likely explains why the specific humidity in the cavity roof is almost always (down to sub-hourly time scales) higher than



in that in the atmosphere despite the relentless mixing by ventilation (moisture surplus relative to the atmosphere; Sect. 5.2), and possibly accounts for the negative July 2022 SEB imbalance (Fig. 10). The upwards vapour transfer in the comparatively large and strongly ventilated cavity during the dry spell might be overly efficient compared with that in the surrounding AL. This might have lead to a non-representatively high specific humidity $q_{sa}$ in the cavity roof and an overestimated evaporative

flux $Q_{LE}$.

### 6.3.4 Aerodynamic roughness lengths

Our mean (median) values of the filtered aerodynamic momentum roughness lengths $z_{0m}$ of 19 to 7 cm (23–8 cm) agree with Mittaz et al. (2000)'s and Stocker-Mittaz (2002)'s values of $18$ and $7$ cm for the snow-free and snow-covered Murtèl surface, respectively (Fig. 9). The scatter range of $\sim 2$ orders of magnitude is similar to that in other studies (Brock et al., 2006; Sicart

et al., 2014; Radić et al., 2017; Fitzpatrick et al., 2017, 2019). The absolute values are at the upper limit of what has been typically found on debris-covered glaciers (Miles et al., 2017) or on Juvvasshøe by Isaksen et al. (2003) ($5$ cm), which is plausible given the rough terrain of the Murtèl rock glacier.

The calculated roughness length decreases slightly with increasing snow height by $0.1$ cm per cm of snow (Fig. 9b). A similar, but much stronger relation between snow height and roughness lengths was found on the Haut Glacier d'Arolla,

where Brock et al. (2006) found $z_{0m}$ to decrease by 2 orders of magnitude with snow heights up to $3$ m. This represents a much stronger relation than that on Murtèl, where $z_{0m}$ decreases by $\sim \times 0.5$ over the observed range of snow thickness. With a maximum snow height of $120$ cm (PERMA-XT measurement; Fig. 3) or $200$ cm (PERMOS) during the measurement period, the snow cover is thin compared with the terrain roughness (edge length of blocks $\sim 50$ cm) and undulations. Both the comparatively high roughness length and the weak sensitivity on snow height might suggest that the aerodynamic roughness is

largely controlled by the furrow-and-ridge micro-topography that is not smoothed out by the snow cover, or by the few largest blocks that stick out of the snow cover rather than by the average block size. In forests, for comparison, the tallest trees of the canopy have a disproportionally large influence on the aerodynamic roughness (Foken, 2017).

For the bulk parametrisation, we linearly interpolate $\log_{10}\{z_{0m}\}$ with snow height. Due to the lack of accurate humidity-corrected sonic temperature measurements, we cannot calculate $z_{0h}$ or $z_{0q}$. We used the unknown ratio $\hat{R}_{z0} := z_{0h}/z_{0m} =$

$z_{0q}/z_{0m}$ as a calibration parameter and found $\hat{R}_{z0} = 2 \times 10^{-2}$. While the approximation $z_{0h} \approx z_{0q}$ seems applicable, $z_{0m} \approx z_{0h}$ is not compatible with our parametrisation (Table 3). This agrees with previous studies on debris-covered glaciers (Steiner et al., 2018, 2021) and hummocky ice surfaces (Smeets and van den Broeke, 2008; Sicart et al., 2014).

### 6.4 Synthesis

The thick coarse-blocky AL strongly insulates the underlying permafrost body. We quantified this well-known effect on Murtèl:

During the thaw seasons 2021 and 2022, roughly $90\%$ of the net radiation $Q^*$ was exported by the turbulent fluxes and not available to melt ground ice (Fig. 13). The ratio between received surface net radiation $Q^*$ to downwards transmitted heat flux $Q_r$ is $Q^* : Q_r \approx 9 : 1$. This follows from the ratio of the measured in-cavity net long-wave radiation $\tilde{Q}_{r_{al}}$ to the surface net radiation $Q^*$ and the relation $\tilde{Q}_{r_{al}} = -0.12 Q^* + 5.6$ W m$^{-2}$ ($R^2 = 0.680$). Our measurements corroborate the relation between



surface net radiation and ground heat flux that has been found by Hoelzle et al. (2022) using PERMOS data from 1997–2019.
Export of the received net radiation is predominantly by sensible fluxes $Q_H$ (50–70%) and secondarily by latent fluxes $Q_{LE}$ (30–50%; Table 2). Surface albedo (spring–early summer snow cover) and sub-surface thermal and moisture regime of the thick coarse-blocky AL control the energy partitioning at/near the surface and the rock glacier's efficiency/ability to export the heat supplied by the surface net radiation $Q^*$, the primary heat input (Fig. 5; Table 2).

More heat was transferred into the ground (and more ground ice observed to melt) during the hot-dry summer 2022 than during the cool-wet summer 2021. Two sets of conditions enhanced the downwards heat transfer $Q_G$ in the hot-dry summer 2022 as measured by the sub-surface pyrgeometer, in particular during dry spells and heat waves: First, less snow in winter 2021–2022 and a warm 2022 spring resulted in an early snow melt in May, one month earlier than in summer 2021, and an early start of the thaw season. This exposed the dark low-albedo blocky surface to the strong insolation and the June heat wave, resulting in a rapid rise of ground temperatures and hence an increase in sensible storage and downwards ground heat flux (Fig. 8; cf. Hoelzle et al. (2022) for a 20-year perspective). Second, the 11-day dry spell in July 2022 exhausted the near-surface moisture stores in the coarse-blocky AL as indicated by the specific humidity deficit $(q_{sa}^* - q_{sa})$ in the cavity roof (Appendix Fig. B2). Lack of moisture limited the evaporative cooling $Q_{LE}$ and countered the rock glacier's ability to export the heat (Fig. 8). The moisture storage capacity of the coarse-blocky Murtèl AL is limited because the near-surface AL presents a rapid drainage and little fine material (silt) to hold water. During dry spells, the evaporation front recedes quickly to greater depths.

Efficient evaporative cooling relies on frequent precipitation events to resupply, as occurred in summer 2021. With climate change, both factors – early start of the thaw season and hot-dry weather spells in summer – are projected to worsen. A shift to earlier snow melting and an increase in the number of snow-free days has already been observed in the Swiss Alps (Hoelzle et al., 2022) and projected to continue with climate change (Scherler et al., 2013; Marmy et al., 2016). Also, the frequency, duration and intensity of heat waves and dry spells are likely to increase. Changes in the SEB of the thermally conditioned rock glaciers and other mountain permafrost landforms entail changes in the ground thermal regime and ground ice content first and, ultimately, morphological changes: thawing, melting of ground ice and degradation.

## 7 Conclusions

We estimated the year-round surface energy balance (SEB) of the seasonally snow-covered ventilated coarse-blocky active layer (AL) of active Murtèl rock glacier that is situated in a cirque in the Upper Engadine (eastern Swiss Alps). The meso-scale landscape produces seasonally contrasting atmospheric conditions of down-slope katabatic jets in winter and a strongly unstable atmosphere in summer. At landform scale, the ventilated near-surface AL acts as a buffer layer where heat and moisture transfer is coupled to the atmosphere, unless sealed by a thick snow cover. Based on a novel sensor array located above ground surface and in the AL that expands an on-site automatic weather station from the Swiss Permafrost Monitoring Network (PERMOS), we were able to improve previous SEB calculations on Murtèl by Mittaz et al. (2000) and Scherler et al. (2014). The measurement period is from September 2020 to September 2022. Our monthly SEB imbalances are within





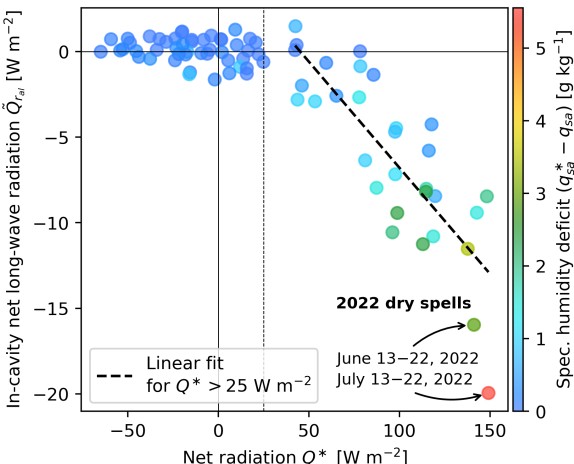

**Figure 13.** Relation between measured 10-day averaged surface net radiation $Q^*$ and measured 10-day averaged in-cavity long-wave radiation $\tilde{Q}_{r_{al}}$ (of the consecutive 10-day window) summarises the relation between $Q^*$, $Q_H$ and $Q_{LE}$. The in-cavity net long-wave radiation $\tilde{Q}_{r_{al}}$ represents the downward heat transfer towards the ground ice table (note that upward heat transfer is primarily by convection and not represented by the radiation measurements; $\tilde{Q}_{r_{al}} \approx 0$ during cooling episodes). Of the available net radiation $Q^*$, 90% is exported into the atmosphere by $Q_H + Q_{LE}$. Only 10% is transmitted deeper into the ground and available to warm the AL and to melt ground ice. Latent turbulent heat export is less efficient during dry spells, when the coarse-blocky AL dries out and evaporation becomes increasingly moisture-limited (outliers from June, July 2022). The relations between in-cavity long-wave radiation and air temperature, surface temperature ($\propto L^{\uparrow}$) or $Q_H$ are qualitatively similar.

20 W m$^{-2}$, except during the snow melt-out months. The main findings concern (i) the climate resilience of the Murtèl rock glacier and (ii) technical aspects of the turbulent flux parametrisations.

(i) Two crucial factors for climate resilience of the Murtèl rock glacier are the insulating high-albedo snow cover and a near-
complete energy turnover during the snow-free thaw season. The two meteorologically contrasting years studied in this work with a cool-wet summer 2021 and a hot-dry summer 2022 were traced into the SEB and ground thermal and moisture regime. First, an early snow melting in 2022 prolonged the thaw season, exposed the dark blocky surface to the intense July insolation and increased AL temperature gradients and the downward heat flux. Second, about 90% of the received surface net radiation was exported and only 10% effectively transferred towards the ground ice table and available to melt ground ice. Heat export
occurs predominantly via sensible turbulent flux and secondarily via the latent turbulent flux from evaporation. The degree of energy turnover and the turbulent flux partitioning is co-controlled by the availability of moisture for evaporation. Dry spells and heat waves counter the rock glacier's ability to export heat by limiting evaporative fluxes leaving the rapidly drying coarse-blocky AL, since the moisture storage capacity is limited. Both trends – earlier snow melt and more heat waves/dry spells – are projected to continue with climate change. This potentially renders coarse-blocky landforms vulnerable to heat waves and
dry spells.



(ii) With our in-mountain permafrost unprecedentedly comprehensive in situ measurements of eddy-covariance flux, liquid precipitation, snow height, AL temperature and humidity profiles, sub-surface airflow speeds and sub-surface long-wave radiation, we tested different bulk parametrisations and constrained their input parameters.

– We successfully parametrised the year-round turbulent fluxes using the modified Louis (1979) scheme (Song, 1998) despite seasonally contrasting atmospheric stability and wind profiles. Wintertime katabatic wind speeds needed to be scaled to close the SEB, which hints at the limits of parametrisations based on the Monin–Obukhov theory in complex mountain terrain and katabatic drainage flows.

– Sensible–latent partitioning of the turbulent fluxes using the simple Bowen ratio approach agreed with the bulk fluxes on a daily–monthly time resolution. Given its parsimony and independence from atmospheric stability and wind speed, the Bowen approach represents a valuable robust tool to estimate the turbulent fluxes in complex terrain with a strongly heterogeneous wind field, provided sub-daily resolution is not required.

– Since daily oscillations of wind speed and insolation are nearly in phase, the sensible turbulent flux $Q_H$ is driven by the gradient between radiometric ground surface temperature $T_s$ and air temperature $T_a$. The flux associated with nocturnal ventilation of the permeable coarse-blocky AL is small due to low wind speeds at night. Using a radiometric surface temperature is convenient for modelling $Q_H$ with remotely sensed data.

– During the thaw season, the evaporative turbulent flux $Q_{LE}$ is 30–50% of the sensible turbulent flux as a function of moisture availability in the near-surface AL. During dry spells, the near-surface moisture stores are exhausted within days, and moisture supply to the surface is limited by the convective vapour transport from the deeper AL.

– Measured eddy-covariance fluxes were systematically too small to close the SEB. Still, the eddy-derived momentum roughness length $z_{0m}$ corroborates previous aerodynamic estimates from Mittaz et al. (2000). $z_{0m}$ varies seasonally between 7 and 20 cm as a function of snow height that smooths the landscape.

*Data availability.* The PERMOS data can be obtained from the PERMOS network (http://www.permos.ch). PERMA-XT data and SEB output are available upon request from the corresponding author.





## Appendix A: Turbulent flux parametrisations

Different formulations of the stability functions exist. Here, we compare the widely used Businger–Dyer relations with the formulation from Louis (1979) and the iterative Monin–Obukhov scheme.

**Businger & Dyer parametrisation**

In the Businger & Dyer parametrisation (denoted by 'cB&D'), the bulk transfer coefficient $C_{bt}$ is expressed as

$$C_{bt} = \frac{k_*^2}{\ln\left\{\frac{z_u}{z_{0m}}\right\}\ln\left\{\frac{z_{T/q}}{z_{0h/q}}\right\}}(\phi_m\phi_{h/q})^{-1} \tag{A1}$$

where the Businger–Dyer non-dimensional stability functions for heat and water vapour are expressed as a function of the bulk Richardson number $\mathrm{Ri}_b$ (Eq. A3) (Brutsaert, 1982; Oke, 1987)

$$(\phi_m\phi_h)^{-1} = (\phi_m\phi_q)^{-1} = \begin{cases} (1 - 5\,\mathrm{Ri}_b)^2, & \text{for } 0.0 \leq \mathrm{Ri}_b \leq 0.2 \text{ (stable case)} \\ (1 - 16\,\mathrm{Ri}_b)^{0.75}, & \text{for } \mathrm{Ri}_b < 0.0 \text{ (unstable case)} \end{cases} \tag{A2}$$

that correct for non-neutral stability of the near-surface atmosphere (atmospheric stratification). These are unity in case of neutrally stable atmosphere ($\mathrm{Ri}_b = 0$), denoted by 'c0'. Used constants and parameters are the von Kármán constant $k_*$ [0.40],

the roughness lengths for momentum $z_{0m}$, heat $z_{0h}$ and water vapour $z_{0q}$, and the (constant) measurement heights for wind speed $z_u$ [m], temperature $z_T$ and humidity $z_q$ above ground surface. In this work, the variable sensor height above the snow surface is accounted for by correcting the wind speed $u$ rather than simply subtracting the snow height from the sensor distance above ground for site-specific reasons discussed in Sect. 6.2.

The Businger–Dyer and the modified Louis (1979) parametrisation characterise atmospheric stability with the bulk Richard-

son number $\mathrm{Ri}_b$ defined by

$$\mathrm{Ri}_b := \frac{g}{\bar{T}}\frac{\frac{T_a - T_s}{z_T - z_{0h}}}{\left(\frac{u}{z_u - z_{0m}}\right)^2} \tag{A3}$$

with the average temperature $\bar{T} := (T_a + T_s)/2$.

However, this widely used stability function (Reid and Brock, 2010; Brock et al., 2010; Mittaz et al., 2000; Favier et al., 2004; Steiner et al., 2018; Scherler et al., 2014) yields implausible fluxes when the atmosphere strongly deviates from near-

neutral stability conditions, $|\mathrm{Ri}_b| > 0.2$. Eq. A2 is invalid for $\mathrm{Ri}_b > +0.2$ and tends to diverge for highly unstable atmospheres at low wind speeds where $\mathrm{Ri}_b \rightarrow -\infty$. Filtering out these fluxes as proposed by Steiner et al. (2018) potentially deletes a sizeable portion of the calculated fluxes at the wind-sheltered Murtèl cirque.

**Modified Louis (1979) scheme**

The Louis (1979) scheme is an analytical approximation of the iterative Monin–Obkuvhov scheme. The bulk exchange factor is

expressed as $C_{bt} = f_h C_{Hn}$ (notation from Essery and Etchevers (2004)), where $C_{Hn} = k_*^2\ln\{z_u/z_0\}^{-2}$ (the neutral exchange



coefficient) and

$$
f_h = \begin{cases} (1 + 10\,\mathrm{Ri}_b)^{-1}, & \text{for } \mathrm{Ri}_b \geq 0 \text{ (stable case)} \\ 1 - 10\,\mathrm{Ri}_b(1 + 10 C_{Hn}\sqrt{-\mathrm{Ri}_b}/f_z)^{-1}, & \text{for } \mathrm{Ri}_b < 0 \text{ (unstable case)} \end{cases} \tag{A4}
$$

with

$$
f_z = \frac{1}{4}\sqrt{\frac{z_0}{z_u}}. \tag{A5}
$$

However, one shortcoming of the Louis (1979) scheme critical on rough surfaces is the assumption of equal momentum
roughness length ($z_{0m}$) and sensible heat roughness length ($z_{0h}$), i.e. $z_{0m} = z_{0h} = z_0$. Hence, we use the Louis (1979) scheme
modified by Song (1998) (denoted by 'cL''). The implementation is described in Rigon et al. (2006); Endrizzi et al. (2014).

**Iterative Monin–Obukhov scheme**

In the Monin–Obkuvhov scheme (denoted by 'cMO'), the atmospheric stability is characterised by the *Obukhov length L*
defined by

$$
L = \frac{u_*^3}{k_* \frac{g}{T_a} \frac{Q_H}{\rho_a C_p}} \tag{A6}
$$

where $u_*\,[\mathrm{m\ s^{-1}}]$ is the friction velocity. The bulk exchange factor for heat $C_{bt} = C_{MO}$ is given by (Conway and Cullen, 2013;
Fitzpatrick et al., 2017; Steiner et al., 2018)

$$
C_{MO} = \frac{k_*^2}{\left[\ln\left\{\frac{z_u}{z_{0m}}\right\} - \psi_m\left(\frac{z_u}{L}\right) + \psi_m\left(\frac{z_{0m}}{L}\right)\right]\left[\ln\left\{\frac{z_T}{z_{0h}}\right\} - \psi_h\left(\frac{z_T}{L}\right) + \psi_h\left(\frac{z_{0h}}{L}\right)\right]} \tag{A7}
$$

where $\psi_m$ and $\psi_h$ are the vertically integrated stability functions for momentum and heat, respectively. The latent flux $Q_{LE}$ is
calculated analogously to scalar roughness length for water vapour $z_{0q}$ and the integrated stability function for water vapour
$\psi_q$, both assumed equal to the corresponding quantity/function for heat. Since $L$ depends itself on $Q_H$, $C_{MO}$ is calculated
iteratively.

Different formulations are available for the integrated stability function $\psi$ as used by e.g. Fitzpatrick et al. (2017); Steiner
et al. (2018); Sauter et al. (2020). We use the following momentum $\psi_m$ and heat $\psi_h$ stability functions (again assuming
$\psi_h = \psi_q$) (Fitzpatrick et al., 2017):

$$
\psi_m(\zeta) = \begin{cases} -\left[a\zeta + b\left(\zeta - \frac{c}{d}\right)\exp\{-d\zeta\} + \frac{bc}{d}\right], & \zeta > 1 \text{ (stable (Beljaars and Holtslag, 1991))} \\ \ln\left\{\left(\frac{1+\chi^2}{2}\right)\left(\frac{1+\chi}{2}\right)^2\right\} - 2\arctan\chi + \frac{\pi}{2}, & \zeta < 0 \text{ (unstable (Dyer, 1974))} \end{cases} \tag{A8}
$$

and the heat stability function $\psi_h$

$$
\psi_h(\zeta) = \begin{cases} -\left[\left(1 + \frac{2a}{3}\zeta\right)^{1.5} + b\left(\zeta - \frac{c}{d}\right)\exp\{-d\zeta\} + \frac{bc}{d} - 1\right], & \zeta > 1 \text{ (stable (Beljaars and Holtslag, 1991))} \\ 2\ln\left(\frac{1+\chi^2}{2}\right), & \zeta < 0 \text{ (unstable (Dyer, 1974))} \end{cases} \tag{A9}
$$

with the dimensionless *Obukhov* parameter $\zeta := z_u/L$ (for $\psi_h$) or $z_T/L$ (for $\psi_h$), $\chi = (1 - 16\zeta)^{1/4}$, $a = 1$, $b = 2/3$, $c = 5$,
$d = 0.35$.



## Appendix B: Ground thermal and hygric regimes

The ground thermal and hygric regimes are shown in Figs. B1, B2. These are the meteorological input variables for the calculation of the turbulent fluxes.

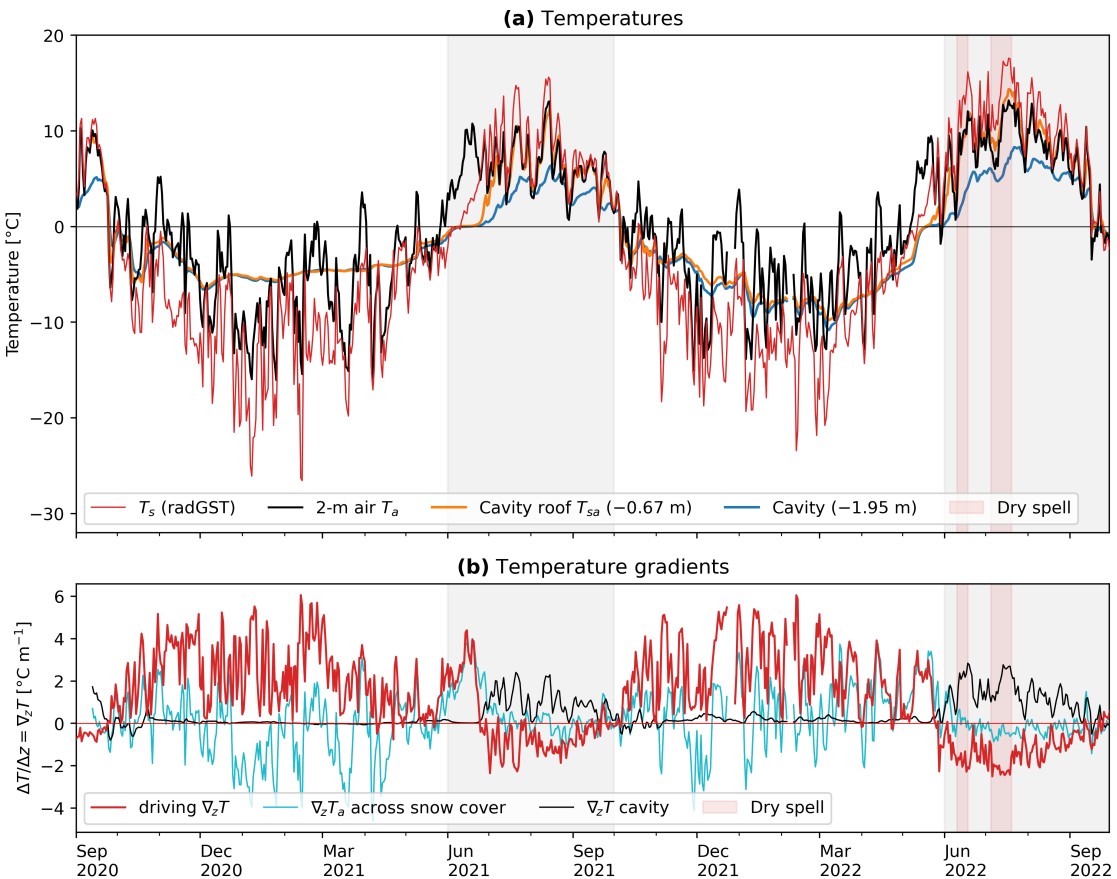

**Figure B1.** Thermal regime. **(a)** Radiometric ground surface temperature $T_s$ ('rGST') and air temperatures in the atmosphere $T_a$ and in the instrumented cavity (cavity roof $T_{sa}$ at 0.67 m and mid-cavity level at 1.95 m depth). **(b):** Vertical temperature gradients $\nabla_z T$ between surface and 2 m air temperature (drives turbulent sensible fluxes; Eq. 11), across the snow cover ($T_a - T_{sa}$) and within the cavity (indicates the stability of the in-cavity air column). The summer 2022 dry spells are referred to in the text.

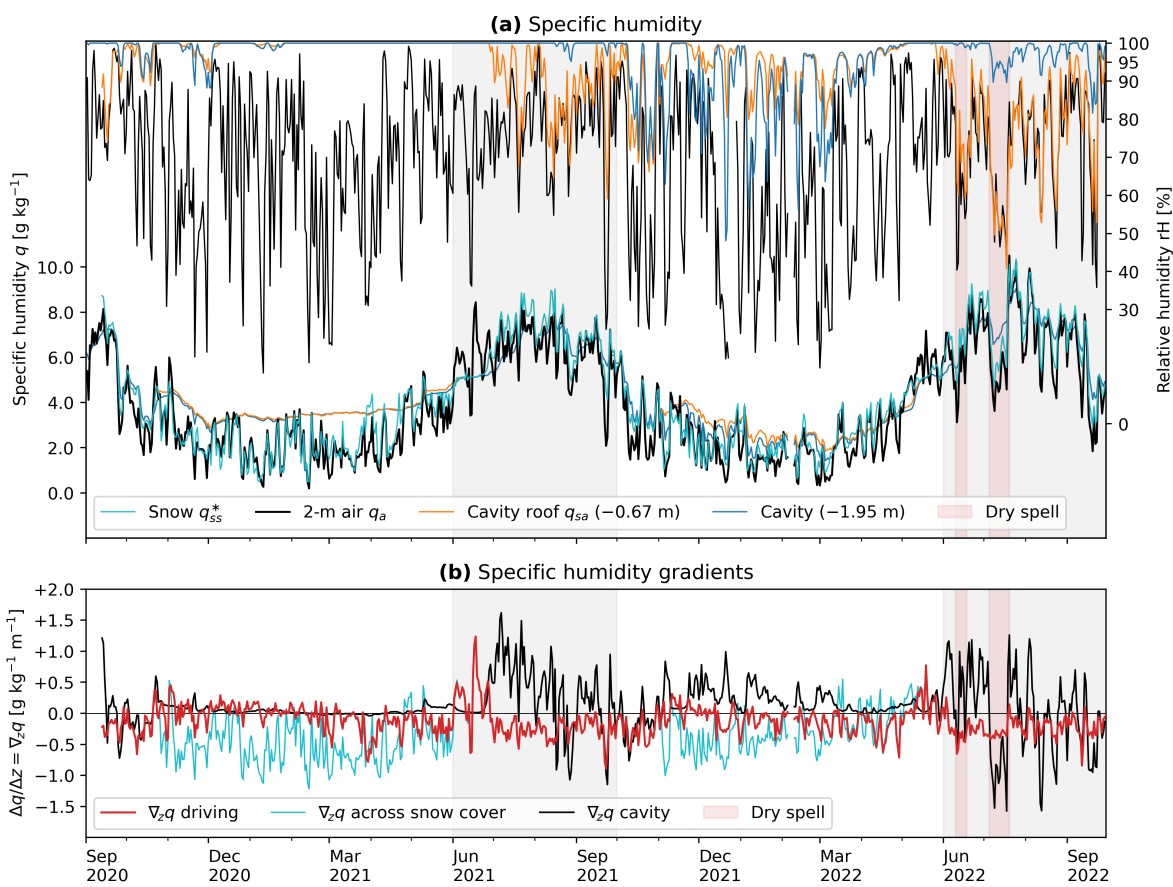

**Figure B2.** Hygric regime. **(a)** Relative and specific humidity of the saturated snow surface $q_{ss}^*$, in the atmosphere $q_a$ and in the instrumented cavity ($q_{sa}$ cavity roof at $0.67$ m and mid-cavity level at $1.95$ m depth). **(b)**: Vertical specific humidity gradients ($\nabla_z q$) between the atmospheric air and the surface (either $q_{ss}^*$ or $q_{sa}$ according to Eq. 10), across the snow cover ($q_a - q_{sa}$) and within the cavity (indicates the moisture transport direction). The moisture transport within the cavity is generally downwards ($\nabla_z q > 0$), and condensation rather than evaporation occurs in the deep AL. Severe dry spells that last long enough to exhaust the near-surface moisture storage ($\sim 5$ days) strongly impact the ground moisture regime, reverse the moisture gradients and lead to upwards moisture transport.

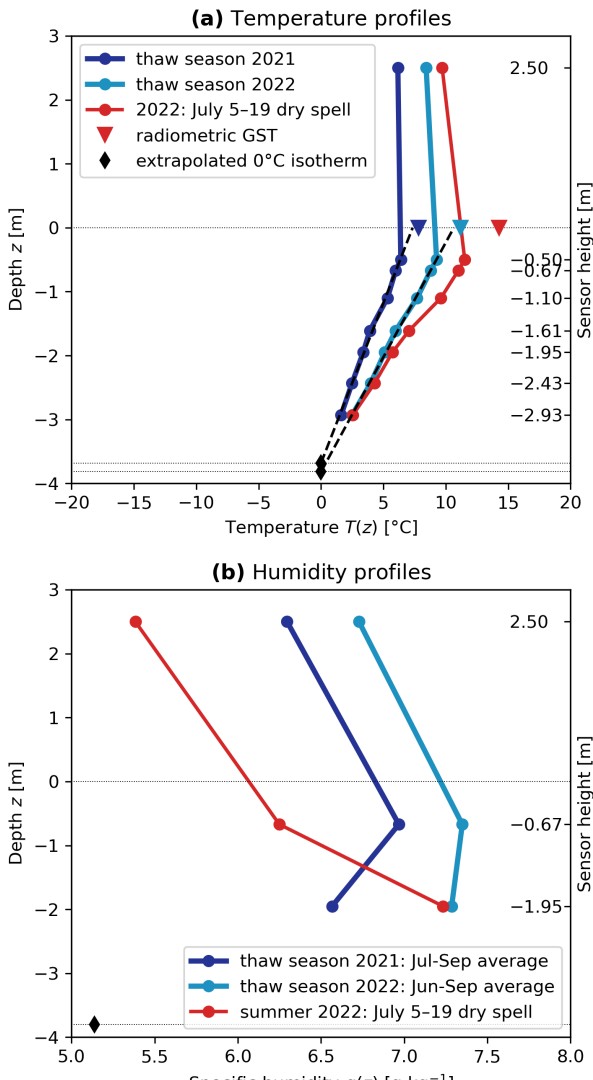

**Figure B3. (a)** Vertical temperature and **(b)** specific humidity profile (thaw-season average). Temperature and humidity are highest in the near-surface AL. The near-surface AL in contact with the atmosphere responds to dry spells and dries out within a few days after the last precipitation event. The deep AL remains close to saturation.



## Appendix C: Estimate of the thermal relaxation time

An estimate for the thermal relaxation time $\Delta t_r$ of a packed bed is

$$\Delta t_r \approx \frac{\rho_r c_r}{h_{\text{eff}}} \frac{V}{A_s} = \frac{\rho_r c_r}{h_{\text{eff}}} \frac{(d_p/2)}{3(1-\phi_d)} \tag{C1}$$

where $h_{\text{eff}}$ is the effective heat transfer coefficient calculated via $1/h_{\text{eff}} = 1/h + d_p/(10\,k_r)$ (Esence et al., 2017). This correction accounts for the additional resistance arising from the temperature gradients within large blocks. 'Large' means blocks whose Biot number exceeds 0.1 (Esence et al., 2017), with the Biot number defined by

$$\text{Bi} := \frac{h}{k_r} L. \tag{C2}$$

The characteristic length $L$ is given by the ratio of the block volume to surface area, $L := V/A_s = d_p/6$ for spheres. A minimum value for the convective heat transfer coefficient $h$ under forced convection at low wind speed ($u < 1\,\text{m s}^{-1}$) is $\sim 10\,\text{W m}^{-2}\,\text{K}^{-1}$. The value is derived from inverting Eq. 11 with $h := Q_H/(T_a - T_s)$ (nighttime values) and agrees with an estimate in Bozhinskiy et al. (1986). With a thermal conductivity of the rock of $k_r = 2.5\,\text{W m}^{-1}\,\text{K}^{-1}$, the critical diameter is $0.15\,\text{m}$. With $d_p = 60\,\text{cm}$ (typical dimension of the blocks enclosing the instrumented cavity), Eq. C1 yields an estimate of $\Delta t_r = \sim 12\,\text{h}$. Thermal adjustment (within $> 95\%$) is attained after $3\Delta t_r$ or $\sim 1.5\,\text{days}$ for large blocks with $60\,\text{cm}$ edge length. Smaller blocks reach thermal equilibrium much faster, e.g. $20\,\text{cm}$ blocks within $10\,\text{h}$.

## Appendix D: 1997–2000 wind speed profiles

Here, we justify our 'katabatic correction' of wind speed measurements with wind profile data collected by Stocker-Mittaz et al. (2002); Stocker-Mittaz (2002). A $10\,\text{m}$ tower installed on the Murtèl rock glacier at the PERMOS monitoring site measured the wind velocity, air temperature and relative humidity at $1.5$, $2.0$, $6.5$ and $9.1\,\text{m}$ above ground surface between January 1997 and March 2000.

Average vertical wind speed profiles (Fig. D1a) show the winter-time low-level jet between December and March with maximum wind speeds close to the snow-covered surface and upwards decreasing wind speed. In summer,wind speeds increase with height and approximately show the log wind profile (Stocker-Mittaz, 2002). The wind speed gradient is predominantly negative for more than $\sim 75\,\text{cm}$ of snow measured at the PERMOS station (located on a plateau) (Fig. D1b), corresponding to $\sim 50\,\text{cm}$ at the PERMA-XT station (located on a more wind-swept ridge). This finding justifies the compensation for snow height (Eq. 23) and possibly explains why the 'katabatic correction' has less effect on the SEB of the snow-poor winter 2021–2022 and a negative effect on the November 2020 SEB (Fig. 10).



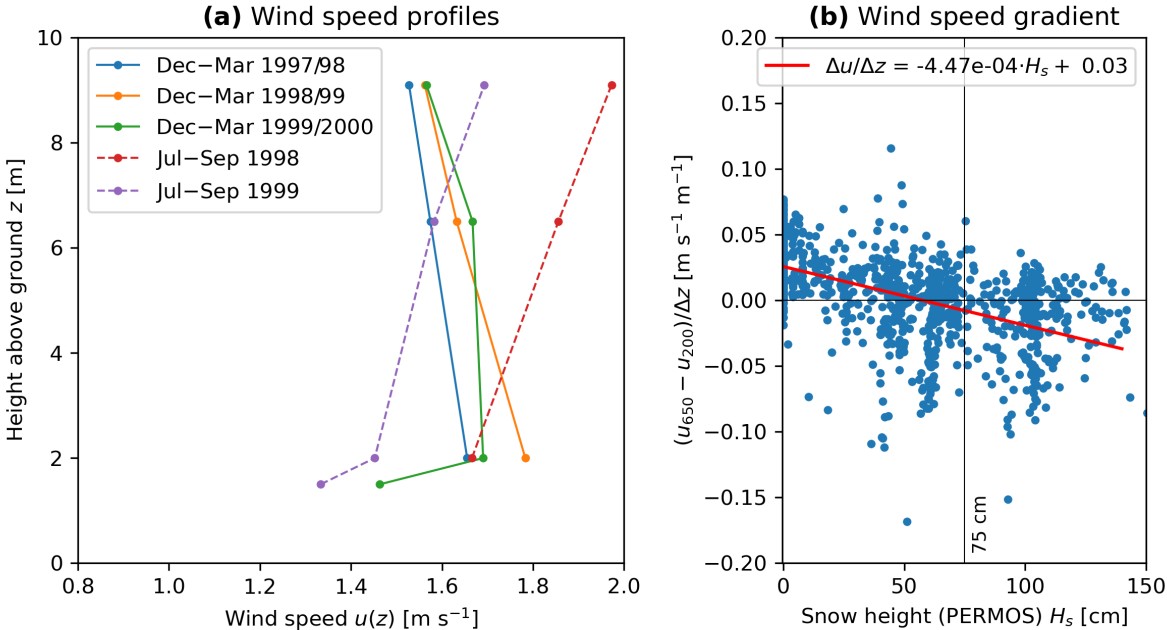

**Figure D1. (a)** Wind speed profiles for winter (December–March average) and summer (July–September) 1997–2000 (Stocker-Mittaz et al., 2002). Winter storms where wind speed at $9.1$ m exceeded $2.5$ m s$^{-1}$ are filtered out (threshold from Stocker-Mittaz (2002). **(b)** The average wind speed gradient between the $2.0$ and $6.5$ m levels switches from a (dominantly) positive log wind profile to a negative low-level jet profile at a (PERMOS) snow height of $\sim 60$ cm ($R^2 = 0.271$). Own figure based on data from Stocker-Mittaz et al. (2002).





## Appendix E: Nomenclature

Parameters and constants used in this study are tabulated in Table E1.

**Table E1.** Nomenclature: Measurement variables, site-specific parameters and constants.

| Symbol | Unit | Name | Symbol | Unit | Name |
|---|---|---|---|---|---|
| $A$ | 1 | Short-wave albedo | $T_r$ | K, °C | Temperature in blocks |
| $C_{bt}$ | 1 | Bulk turbulent heat and vapour transfer coefficient | $t$ | s, h | Time |
| $C_p$ | J kg$^{-1}$ K$^{-1}$ | Isobaric specific heat capacity of moist air | $u$ | m s$^{-1}$ | Wind or airflow speed |
| $C_w, C_r$ | J m$^{-3}$ K$^{-1}$ | Volumetric heat capacity of water, rock | $u_*$ | m s$^{-1}$ | Friction velocity |
| $e, e^*$ | Pa | Vapour pressure (at saturation) | $\dot{W}_{rain}$ | m$^3$ s$^{-1}$ m$^{-2}$ | Rainfall rate |
| $K_h,$ $K_w$ | m$^2$ s$^{-1}$ | Eddy diffusivity for sensible and latent heat | $z_u, z_T,$ $z_q$ | m | Measurement height of wind speed, air temperature and humidity |
| $k_s$ | W m$^{-1}$ K$^{-1}$ | Thermal conductivity of the snow cover | $z_{0m},$ $z_{0h}, z_{0q}$ | m | Roughness length for momentum, heat and water vapour |
| $H_d, H_s$ | m | Thickness of coarse-blocky AL, snow cover | $z$ | m | Vertical coordinate |
| $^{\mathrm{SEB}}H_s^{\mathrm{crit}}$ | m | Thickness of decoupling snow cover | Bo | 1 | Bowen ratio, Bo $:= Q_H/Q_{LE}$ |
| $\hat{m}$ | 1 | 'Katabatic correction' exponent (calibration parameter) | Ri$_b$ | 1 | Bulk Richardson number (Eq. A3) |
| $P$ | Pa | Atmospheric pressure | $\varepsilon$ | 1 | Surface emissivity (snow, blocky surface) |
| $Q, S, L$ | W m$^{-2}$ | Heat flux (general, short-wave radiation, long-wave radiation) | $\bar{\rho}_s, \rho_a,$ $\rho_r$ | kg m$^{-3}$ | Density of the snowpack, air and rock |
| $q, q^*$ | g g$^{-1}$ | Specific humidity (at saturation) | $\phi_d$ | 1 | Porosity of coarse-blocky AL |
| $q_a, q_s$ | g g$^{-1}$ | Specific humidity (air 2 m above ground, at surface) | $\phi_m, \phi_h,$ $\phi_q$ | 1 | Stability functions for momentum, heat and water vapour |
| $q_{ss}^*$ | g g$^{-1}$ | Saturated specific humidity of the snow surface | $\psi_m,$ $\psi_h, \psi_q$ | 1 | Integrated stability functions |
| $q_{sa}$ | g g$^{-1}$ | Specific humidity in the near-surface AL | $C_{pd}$ | J kg$^{-1}$ K$^{-1}$ | Heat capacity of dry air (1005) |
| $\hat{R}_{z0}$ | 1 | Ratio of momentum and scalar roughness lengths (calibration parameter) | $g$ | m s$^{-2}$ | Gravitational acceleration (9.81) |
| rH | % | Relative humidity | $k_*$ | 1 | von Kármán constant (0.4) |
| SWE | kg m$^{-2}$ | Snow water equivalent | $L_m$ | J kg$^{-1}$ | Latent heat of melting ($3.36 \times 10^5$) |
| $T_s$ | K, °C | Surface temperature (coarse-blocky AL, snow surface) | $L_s$ | J kg$^{-1}$ | Latent heat of sublimation ($2.83 \times 10^6$) |
| $T_b$ | K, °C | Temperature at base of snow cover | $L_v$ | J kg$^{-1}$ | Latent heat of vaporisation ($2.476 \times 10^6$) |
| $T_a,$ $T_{wb}, T_v$ | K, °C | Air temperature (dry-bulb, wet-bulb, virtual) | $\sigma$ | W m$^{-2}$ K$^{-4}$ | Stefan–Boltzmann constant ($5.670 \times 10^{-8}$) |



*Author contributions.* DA performed the fieldwork, model development and analyses for the study and wrote the manuscript. MS, MH and BK supervised the study, provided financial and field support and contributed to the manuscript preparation. AH and CK provided logistical
support and editorial suggestions on the manuscript. HG designed the novel sensor array, regularly checked data quality, contributed to the analyses and provided editorial suggestions on the manuscript.

*Competing interests.* The authors declare that they have no conflict of interest.

*Acknowledgements.* This work is a collaboration between the University of Fribourg and GEOTEST and was funded by the Swiss Innovation Agency InnoSuisse (project 36242.1 IP-EE 'Permafrost Meltwater Assessment eXpert Tool PERMA-XT'). The authors wish to thank Walter
Jäger (Waljag GmbH, Malans) and Thomas Sarbach (Sarbach Mechanik, St. Niklaus) for the technical support, and the Corvatsch cable car company for logistical support. This publication is dedicated to Martin Scherler who laid the conceptual foundation.



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
