# Peer review of "Surface heat fluxes at coarse-blocky Murtèl rock glacier (Engadine, eastern Swiss Alps)"

_EGUsphere, 2023_

## Author Comment (AC1)

Response to the first referee comments to "Surface heat fluxes at coarse-blocky Murtel rock glacier (Engadine, eastern Swiss Alps)" by Amschwand et al. (author comments)

*Reviewer's comments are shown in black italics.* Responses from the authors are presented in blue regular font below each comment. Citations from the manuscript are in Times New Roman, changes of the manuscript text are underlined.
December 10, 2023

RC1: 'Comment on egusphere-2023-2109', Anonymous Referee #1, 29 Oct 2023

*The authors attempt to describe the surface energy balance in the active layer of a rock glacier in the Swiss Alps with an impressively well thought out measurement setup within a cavity complemented by a well established monitoring setup of the atmosphere above. Over two years they show how the energy input is nearly completely turned over, explaining the resilience rock glaciers have versus potential drivers of mass wasting but also providing an understanding what parameters are crucial to keep in mind in future, to anticipate the potential decay of ground ice in such landforms under a changing climate. They are also able to test a variety of stability corrections and discuss their suitability to estimate turbulent fluxes efficiently and show suitable fits for already established models for a rock glacier case.*
*The study is extremely well thought out, relies on an impressive instrumentation array and both methods and results are described in lucid language. As much as I would like to get further into a discussion of individual details of the study, it is already so dense and well crafted that I can not propose any big changes before I would recommend this for publication. I believe this work will be a crucial contribution to the community's understanding of potential resilience of rock glaciers (and even debris covered ones) as well as when and how this resilience may come to an end. Future studies will hugely benefit from the ideas in monitoring and some of the concepts proposed in the Results to further this understanding. I am looking forward to this being part of future discussions on the topic.*

*There are very few general comments I have below and even fewer minor ones, as the authors have done an excellent job in language, syntax and copy editing already.*
We thank reviewer #1 for this enthusiastically positive and encouraging assessment as well as for the careful reading. We address the raised points below and hope that we can positively answer them.

*General:*

*L112: It is unclear what you mean with 'massive ice', please clarify. You mean depth? Or high content?*
We hope to clarify the text as follows: Drill cores have revealed sparsely sand- and silt-bearing massive ice (3–28 m depth, ice content over 90% by volume).

*L130ff: Definitely important to point out the crucial role this laboratory has, I am just wondering if you could reduce the list of citations here to a few key publications that anyway then point to these further papers (e.g. 'Hoelzle et al. 2002 and Scherler et al. 2014 and references therein, as well as at least 10 unpublished MSc theses'). The theses can not be directly accessed so am not sure how useful to cite here (or provide links to Uni repositories if possible). Also in references you call them 'phdthesis' and 'mathesis', if you keep them write it out so it is clear (PhD Thesis, MSc Thesis). Also you switch between 'Umlaut' spellings (e.g. 'Ueber den Wärmehaushalt…'), keep it uniform.*
Unfortunately, most of these MSc theses are not digitised and not available online. We will modify the text as suggested.

*L146: If reported in both enough if you just refer to one here.*
We have chosen one reference.

*L445: It wasn't completely clear to me what 'time period bias' is supposed to mean. Can you explain that further?*

We take daily temperature amplitudes in the near-surface active layer as a proxy of the degree of ventilation and convective coupling: similar amplitudes mean coupled; strongly attenuated amplitudes mean decoupled. The data as presented in the paper draft (Fig. 4) might suggest that decoupling begins sharply at 20 cm of snow. However, the WS wind speed measurements suggest rather a gradual decoupling than a sharp one. We think that the 'gap' in Fig. 4 marked by the `??' appears because snow depths between 20 and 40 cm were rarely observed in the study period 2020-2022. A few major snowfall events made the snow height rise from 20 to 40 cm within hours. Adding new data from the snow-poor winter 2022-2023 (even less snow than 2021-2022) does fill the gap (Figure below). In the revised version, we'll remove the misleading phrase `time period bias' and explain it more clearly.

[Figure]

*L713: I am surprised this happens only 'occasionally' (Ts < Ta) – could you specify on how many nights or how many hourly measurements fulfilled that criterion? On debris cover we see this reversal relatively frequently.*

Yes, this reversal occurs during clear sky nights as a rule, and clear-sky nights occur frequently in the Engadine valley (e.g., August 2022: 24 out of 31 nights). These reversals are common, as reviewer #1 suspected. We will delete the word 'occasionally' and emphasize its frequency in the dry Engadine valley.

*L839: maybe better 'daily as well as monthly'?*

We have modified the text to: Sensible--latent partitioning of the turbulent fluxes using the simple Bowen ratio approach agreed with the bulk fluxes on monthly down to daily resolution.

*L851: I understand that the data was extremely difficult to retrieve and is valuable but I would invite the authors to consider making it available through a repository in the spirit of open science. I believe no scientist without the detailed field knowledge and knowledge of the setup would be able to take away any of your future research ideas before you are able to execute them. Conversely I believe that this dataset could be an extremely crucial contribution to our better understanding of cooling (or the future end of it) in such surface covers, and giving the opportunity to others who have experience in such environments to think along with this data, could accelerate our understanding even further.*

We agree and have prepared the publication of the data set. It will be available on the PERMOS data portal, https://www.permos.ch/data-portal/data-publications-doi .

*Minor:*

*L44: '… exceptions being Rist and Phillips (2005) and Rist (2007)'*
We have modified the text accordingly.

*L85: You use the 'sensu Oke' reference already in L35, I think here it is redundant as this is already introduced.*
We agree and have removed the reference.

*L330: You mean 'could not (!) test this bulk method'?*
Thank you for catching this error. We have added the (rather important) "not" in the sentence.

*Citation: https://doi.org/10.5194/egusphere-2023-2109-RC1*
* * *
RC2: 'Comment on egusphere-2023-2109', Anonymous Referee #2, 31 Oct 2023

*This manuscript presents a thorough analysis based on a substantial amount of field data. As such, it is not an easy read, but I cannot identify anything that I would suggest cutting (there is an amount of repetition that could be removed). The statements that the parametrisation "closed the monthly SEB" but "wind speeds had to be scaled to close the SEB" using a calibration parameter are alarming, but my comments are otherwise minor.*
We thank reviewer #2 for this positive assessment as well as for the careful reading (notably of the equations). We understand the concerns about the turbulent flux parameterisations for the winter-time katabatic winds and agree that this is a shortcoming of this study – a shortcoming however that is known (Grisogono et al., 2007) and that we cannot properly address with our one-level wind-speed measurements. We opted for a short ad hoc solution, which we name as such, show the SEB deviation with and without this correction for traceability (Fig. 10), show wind-speed profiles of the katabatic jet from previous research (Fig. D1; Stocker-Mittaz et al., 2002), and give a suggestion for improvement (Oerlemans and Grisogono (2002)'s katabatic model).

*Figure 2*
*Photograph not mentioned in the caption.*
We have clarified the figure by assigning panel (b) to the photo and mentioned in the caption.

*181*
*Could say that the Sicart et al. (2005) correction is for interference of solar radiation.*
We have modified the text accordingly: (3) Sicart correction of the incoming long-wave radiation for interference with solar radiation.

*214*
*The psychrometric constant here is the theoretical value for a ventilated wet bulb. Is that appropriate?*
We use the wet-bulb temperature to discriminate the precipitation phase, which in dry conditions as at our study site works better than the dry-bulb air temperature (Froidurot et al., 2014). We think that the psychrometric constant is sufficient for our purpose of getting the frequency of solid or mixed precipitation. No calculations are based on the wet-bulb temperature.

*272*
*Missing ) in $C_p$*
Thank you for catching this typo. We have added the missing parenthesis.

*330*
*"could not test this bulk method"*
Thank you for catching this error. We have added the (rather important) "not" in the sentence.

*379*

*The modulus for downwards longwave radiation is unnecessary.*

We agree (as measured radiation is always positive) and have removed the modulus. Additionally, we have moved this Eq. 21 into Sect. 3.25 where it is more appropriate.

*Figure 7*

*First reference to this figure in the text comes after Fig. 11.*

*The black points on top of blue points are fine on screen but did not print well.*

We'll reverse the order of Figs. 7 and 11, and make black points better visible by drawing a white border.

*Figure 12*

*The "Air flow speed proxy" axis needs an upper value.*

*The caption or legend should state that the dashed lines are latent heat fluxes.*

We have modified the plot (new version shown below), added the legend for latent heat fluxes (panel d), and added axis ticks (panel c). Furthermore, we put the active layer (AL) airflow speed into its own panel (c) to avoid the impression that airflow speed in the AL is as strong as the 2-m wind speed.

[Figure]

*757*

*The use of parentheses and "respectively" make for a mangled sentence.*

We have disent(m)angled the sentence: The filtered aerodynamic momentum roughness lengths $z_{0m}$ depend on the snow height. Our mean value of 19 cm agrees with Stocker-Mittaz (2002)'s value of

18 cm for snow-free conditions, and 7 cm for snow-covered conditions. The median values are slightly higher, 23 cm for snow-free and 8 cm for snow-covered conditions.

*872*
*Having given Ta as being measured in degrees Celsius in Table 1, need to state that this is a Kelvin temperature.*
Thank you for catching the missing unit. We have modified the text accordingly.

*905*
*The first \psi_h should be \psi_m*
Thank you for catching this typo. We have modified the text accordingly.

*Citation: https://doi.org/10.5194/egusphere-2023-2109-RC2*

References

Froidurot, S., Zin, I., Hingray, B., and Gautheron, A.: Sensitivity of Precipitation Phase over the Swiss Alps to Different Meteorological Variables, Journal of Hydrometeorology, 15, 685–696, https://doi.org/10.1175/JHM-D-13-073.1, 2014.

Grisogono, B., Kraljević, L., and Jeričević, A.: The low-level katabatic jet height versus Monin–Obukhov height, Quarterly Journal of the Royal Meteorological Society, 133, 2133–2136, https://doi.org/10.1002/qj.190, 2007.

Oerlemans, J. and Grisogono, B.: Glacier winds and parameterisation of the related surface heat fluxes, Tellus A: Dynamic Meteorology and Oceanography, 54, 440, https://doi.org/10.3402/tellusa.v54i5.12164, 2002.

Stocker-Mittaz, C., Hoelzle, M., and Haeberli, W.: Modelling alpine permafrost distribution based on energy-balance data: a first step, Permafrost and Periglacial Processes, 13, 271–282, https://doi.org/10.1002/ppp.426, 2002.

---

## Author Response (AR1)

Response to the first referee comments to "Surface heat fluxes at coarse-blocky Murtèl rock glacier (Engadine, eastern Swiss Alps)" by Amschwand et al. (author comments)

*Reviewer's comments are shown in black italics.* Responses from the authors are presented in blue regular font below each comment. Citations from the manuscript are in Times New Roman, changes of the manuscript text are underlined.
January 5, 2024

RC1: 'Comment on egusphere-2023-2109', Anonymous Referee #1, 29 Oct 2023

*The authors attempt to describe the surface energy balance in the active layer of a rock glacier in the Swiss Alps with an impressively well thought out measurement setup within a cavity complemented by a well established monitoring setup of the atmosphere above. Over two years they show how the energy input is nearly completely turned over, explaining the resilience rock glaciers have versus potential drivers of mass wasting but also providing an understanding what parameters are crucial to keep in mind in future, to anticipate the potential decay of ground ice in such landforms under a changing climate. They are also able to test a variety of stability corrections and discuss their suitability to estimate turbulent fluxes efficiently and show suitable fits for already established models for a rock glacier case.*
*The study is extremely well thought out, relies on an impressive instrumentation array and both methods and results are described in lucid language. As much as I would like to get further into a discussion of individual details of the study, it is already so dense and well crafted that I can not propose any big changes before I would recommend this for publication. I believe this work will be a crucial contribution to the community's understanding of potential resilience of rock glaciers (and even debris covered ones) as well as when and how this resilience may come to an end. Future studies will hugely benefit from the ideas in monitoring and some of the concepts proposed in the Results to further this understanding. I am looking forward to this being part of future discussions on the topic.*

*There are very few general comments I have below and even fewer minor ones, as the authors have done an excellent job in language, syntax and copy editing already.*
We thank reviewer #1 for this enthusiastically positive and encouraging assessment as well as for the careful reading. We address the raised points below and hope that we can positively answer them.

*General:*

*L112: It is unclear what you mean with 'massive ice', please clarify. You mean depth? Or high content?*
We have clarified the text as follows: Drill cores have revealed sparsely sand- and silt-bearing massive ice (3–28 m depth, ice content over 90% by volume).

*L130ff: Definitely important to point out the crucial role this laboratory has, I am just wondering if you could reduce the list of citations here to a few key publications that anyway then point to these further papers (e.g. 'Hoelzle et al. 2002 and Scherler et al. 2014 and references therein, as well as at least 10 unpublished MSc theses'). The theses can not be directly accessed so am not sure how useful to cite here (or provide links to Uni repositories if possible). Also in references you call them 'phdthesis' and 'mathesis', if you keep them write it out so it is clear (PhD Thesis, MSc Thesis). Also you switch between 'Umlaut' spellings (e.g. 'Ueber den Wärmehaushalt…'), keep it uniform.*
Unfortunately, most of these MSc theses are not digitised and not available online. We have deleted these references.

*L146: If reported in both enough if you just refer to one here.*
We have chosen one reference.

*L445: It wasn't completely clear to me what 'time period bias' is supposed to mean. Can you explain that further?*

We have removed the misleading phrase `time period bias' and explained it more clearly with additional winter 2022-2023 data:

Normalised near-surface AL temperature amplitudes indicate the degree of convective coupling: amplitudes similar to that in the atmosphere means coupled; strongly attenuated amplitudes means decoupled (Fig. 4a; additional winter 2022-2023 data shown). The decoupling proceeded gradually with snow depth (sketched by the schematic envelope), the thresholds are only approximate values.

*L713: I am surprised this happens only 'occasionally' (Ts < Ta) – could you specify on how many nights or how many hourly measurements fulfilled that criterion? On debris cover we see this reversal relatively frequently.*

These reversals are common, as reviewer #1 suspected. We have changed the text to:
Furthermore, the measured eddy-covariance flux remained upwards-directed in the early morning hours, when the terrain surface has radiatively cooled to air temperature $T_s \approx T_a$, or even during reversals $T_s < T_a$ in clear-sky nights (that frequently occur in the Engadine: e.g., 24 out of 31 nights in August 2022).

*L839: maybe better 'daily as well as monthly'?*

We have changed the text to: Sensible-latent partitioning of the turbulent fluxes using the simple Bowen ratio approach agreed with the bulk fluxes on monthly to daily resolution.

*L851: I understand that the data was extremely difficult to retrieve and is valuable but I would invite the authors to consider making it available through a repository in the spirit of open science. I believe no scientist without the detailed field knowledge and knowledge of the setup would be able to take away any of your future research ideas before you are able to execute them. Conversely I believe that this dataset could be an extremely crucial contribution to our better understanding of cooling (or the future end of it) in such surface covers, and giving the opportunity to others who have experience in such environments to think along with this data, could accelerate our understanding even further.*

We agree and have published the measurement data set. It is available on the PERMOS data portal via https://www.permos.ch//doi/permos-spec-2023-1 (doi:10.13093/permos-spec-2023-01).

*Minor:*

*L44: '… exceptions being Rist and Phillips (2005) and Rist (2007)'*

We have modified the text accordingly.

*L85: You use the 'sensu Oke' reference already in L35, I think here it is redundant as this is already introduced.*

We have removed the reference.

*L330: You mean 'could not (!) test this bulk method'?*

Thank you for catching this error. We have added the (rather important) "not" in the sentence.

*Citation: https://doi.org/10.5194/egusphere-2023-2109-RC1*
* * *
RC2: 'Comment on egusphere-2023-2109', Anonymous Referee #2, 31 Oct 2023

*This manuscript presents a thorough analysis based on a substantial amount of field data. As such, it is not an easy read, but I cannot identify anything that I would suggest cutting (there is an amount of*

*repetition that could be removed). The statements that the parametrisation "closed the monthly SEB" but "wind speeds had to be scaled to close the SEB" using a calibration parameter are alarming, but my comments are otherwise minor.*

We thank reviewer #2 for this positive assessment as well as for the careful reading (notably of the equations). We understand the concerns about the turbulent flux parameterisations for the winter-time katabatic winds and agree that this is a shortcoming of this study – a shortcoming however that is known (Grisogono et al., 2007) and that we cannot properly address with our one-level wind-speed measurements. We opted for a short ad hoc solution, which we name as such, show the SEB deviation with and without this correction for traceability (Fig. 10), show wind-speed profiles of the katabatic jet from previous research (Fig. D1; Stocker-Mittaz et al., 2002), and give a suggestion for improvement (Oerlemans and Grisogono (2002)'s katabatic model).

We state that in the abstract more clearly as:

The monthly SEB is closed within 20 W m$^{-2}$, except during the snowmelt months and under katabatic drainage winds in winter.

*Figure 2*
*Photograph not mentioned in the caption.*

We have clarified the figure by assigning panel (b) to the photo and mentioned in the caption.

*181*
*Could say that the Sicart et al. (2005) correction is for interference of solar radiation.*

We have modified the text accordingly: (3) Sicart correction of the incoming long-wave radiation for interference with solar radiation.

*214*
*The psychrometric constant here is the theoretical value for a ventilated wet bulb. Is that appropriate?*

We use the wet-bulb temperature to discriminate the precipitation phase, which in dry conditions as at our study site works better than the dry-bulb air temperature (Froidurot et al., 2014). We think that the psychrometric constant is sufficient for our purpose of getting the frequency of solid or mixed precipitation. No calculations are based on the wet-bulb temperature.

*272*
*Missing ) in C_p*

Thank you for catching this typo. We have added the missing parenthesis.

*330*
*"could not test this bulk method"*

Thank you for catching this error. We have added the (rather important) "not" in the sentence.

*379*
*The modulus for downwards longwave radiation is unnecessary.*

We agree (as measured radiation is always positive) and have removed the modulus. Additionally, we have moved this Eq. 21 into Sect. 3.25 where it is more appropriate.

*Figure 7*
*First reference to this figure in the text comes after Fig. 11.*
*The black points on top of blue points are fine on screen but did not print well.*

We have reversed the order of Figs. 7 and 11, and have drawn the blue/orange points slightly transparent to make the black points better visible.

*Figure 12*

*The "Air flow speed proxy" axis needs an upper value.*

*The caption or legend should state that the dashed lines are latent heat fluxes.*

We have modified the plot (new version shown below), added the legend for latent heat fluxes (panel d), and added axis ticks (panel c). Furthermore, we put the active layer (AL) airflow speed into its own panel (c) to avoid the impression that airflow speed in the AL is as strong as the 2-m wind speed.

[Figure]

*757*

*The use of parentheses and "respectively" make for a mangled sentence.*

We have disent(m)angled the sentence: The filtered aerodynamic momentum roughness lengths $z_{0m}$ depend on the snow height. Our mean value of 19 cm agrees with Stocker-Mittaz (2002)'s value of 18 cm for snow-free conditions, and 7 cm for snow-covered conditions. The median values are slightly higher, 23 cm for snow-free and 8 cm for snow-covered conditions.

*872*

*Having given Ta as being measured in degrees Celsius in Table 1, need to state that this is a Kelvin temperature.*

Thank you for catching the missing unit. We have modified the text accordingly.

*905*

*The first \psi_h should be \psi_m*

Thank you for catching this typo. We have modified the text accordingly.

***Citation**: https://doi.org/10.5194/egusphere-2023-2109-RC2*

Additional minor changes to the manuscript (not affecting the content or message):

- Clarified the notation by renaming the sensible heat storage changes $\Delta Q_{stor}$ to $\Delta H^\theta$ and $\Delta H_S$ to $\Delta H_S$; fluxes and storage are more clearly named. Snow depth was renamed from $H_S$ to $h_S$.
- Updated Fig. 4 with winter 2022/2023 snow height.

References

Froidurot, S., Zin, I., Hingray, B., and Gautheron, A.: Sensitivity of Precipitation Phase over the Swiss Alps to Different Meteorological Variables, Journal of Hydrometeorology, 15, 685–696, https://doi.org/10.1175/JHM-D-13-073.1, 2014.

Grisogono, B., Kraljević, L., and Jeričević, A.: The low-level katabatic jet height versus Monin–Obukhov height, Quarterly Journal of the Royal Meteorological Society, 133, 2133–2136, https://doi.org/10.1002/qj.190, 2007.

Oerlemans, J. and Grisogono, B.: Glacier winds and parameterisation of the related surface heat fluxes, Tellus A: Dynamic Meteorology and Oceanography, 54, 440, https://doi.org/10.3402/tellusa.v54i5.12164, 2002.

Stocker-Mittaz, C., Hoelzle, M., and Haeberli, W.: Modelling alpine permafrost distribution based on energy-balance data: a first step, Permafrost and Periglacial Processes, 13, 271–282, https://doi.org/10.1002/ppp.426, 2002.